# The Tunnel Effect: Building Data Representations in Deep Neural Networks

**Wojciech Masarczyk[1],***    **Mateusz Ostaszewski[1]**    **Ehsan Imani[2]**    **Razvan Pascanu[3]**

**Piotr Miłoś[4,5]**        **Tomasz Trzciński[1,4,6]**

## Abstract

Deep neural networks are widely known for their remarkable effectiveness across various tasks, with the consensus that deeper networks implicitly learn more complex data representations. This paper shows that sufficiently deep networks trained for supervised image classification split into two distinct parts that contribute to the resulting data representations differently. The initial layers create linearly-separable representations, while the subsequent layers, which we refer to as *the tunnel*, compress these representations and have a minimal impact on the overall performance. We explore the tunnel's behavior through comprehensive empirical studies, highlighting that it emerges early in the training process. Its depth depends on the relation between the network's capacity and task complexity. Furthermore, we show that the tunnel degrades out-of-distribution generalization and discuss its implications for continual learning.

## 1 Introduction

Neural networks have been the powerhouse of machine learning in the last decade. A significant effort has been put into understanding the mechanisms underlying their effectiveness. One example is the analysis of building representations in neural networks applied to image processing [1]. The consensus is that networks learn to use layers in the hierarchy by extracting more complex features than the layers before [2, 3], meaning that each layer contributes to the final network performance.

Extensive research has shown that increasing network depth exponentially enhances

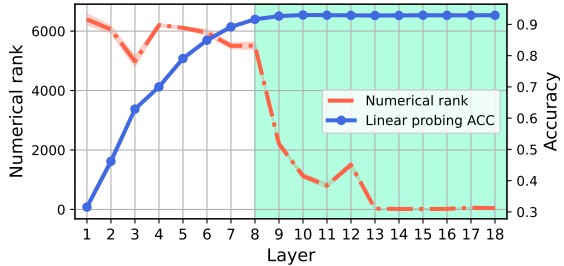

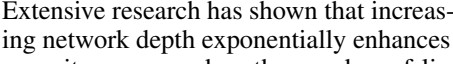

Figure 1: **The tunnel effect** for VGG19 trained on the CIFAR-10. In the tunnel (shaded area), the performance of linear probes attached to each layer saturates (blue line), and the representations rank is steeply reduced (red dashed line).

capacity, measured as the number of linear regions [4–6]. However, practical scenarios reveal that deep and overparameterized neural networks tend to simplify representations with increasing

---

[1]Warsaw University of Technology, Poland

[2]University of Alberta, Canada

[3]University College London, UK

[4]IDEAS NCBR, Poland

[5]Institute of Mathematics, Polish Academy of Sciences, Poland

[6]Tooploox, Poland

*Corresponding author: `wojciech.masarczyk@gmail.com`

37th Conference on Neural Information Processing Systems (NeurIPS 2023).

depth [7, 8]. This phenomenon arises because, despite their large capacity, these networks strive to reduce dimensionality and focus on discriminative patterns during supervised training [7–10]. Motivated by these findings, we aim to investigate this phenomenon further and formulate the following research question:

*How do representations depend on the depth of a layer?*

Our investigation focuses on severely overparameterized neural networks through the prism of their representations as the core components for studying neural network behavior [11, 12].

We extend the commonly held intuition that deeper layers are responsible for capturing more complex and task-specific features [13, 14] by showing that neural networks after learning low and high-level features use the remaining layers for compressing obtained representations.

Specifically, we demonstrate that deep neural networks split into two parts exhibiting distinct behavior. The first part, which we call the extractor, builds representations, while the other, dubbed *the tunnel*, propagates the representations further to the model's output, compressing them significantly. As we show, this behavior has important implications for generalization, transfer learning, and continual learning. To investigate the tunnel effect, we conduct multiple experiments that support our findings and shed some light on the potential source of this behavior. Our findings can be summarized as follows:

- We conceptualize and extensively examine the tunnel effect, namely, deep networks naturally split into *the extractor* responsible for building representations and *the compressing tunnel*, which minimally contributes to the final performance. The extractor-tunnel split emerges early in training and persists later on.
- We show that the tunnel deteriorates the generalization ability on out-of-distribution data.
- We show that the tunnel exhibits task-agnostic behavior in a continual learning scenario. Simultaneously it leads to higher catastrophic forgetting of the model.

## 2   The tunnel effect

The paper introduces and studies the dynamics of representation building in overparameterized deep neural networks called *the tunnel effect*. The following section validates the tunnel effect hypothesis in a number of settings. Through an in-depth examination in Section 3.1, we reveal that the tunnel effect is present from the initial stages and persists throughout the training process. Section 3.2 focuses on the out-of-distribution generalization and representations compression. Section 3.3 hints at important factors that impact the depth of the tunnel. Finally, in Section 4, we confront an auxiliary question: How does the tunnel's existence impact a model's adaptability to changing tasks and its vulnerability to catastrophic forgetting? To answer these questions we formulate our main claim as:

***The tunnel effect hypothesis:*** *Sufficiently large * neural networks develop a configuration in which network layers split into two distinct groups. The first one which we call the extractor, builds linearly-separable representations. The second one, the tunnel, compresses these representations, hindering the model's out-of-distribution generalization.*

### 2.1   Experimental setup

To examine the phenomenon, we designed the setup to include the most common architectures and datasets, and use several different metrics to validate the observations.

**Architectures** We use three different families of architectures: MLP, VGGs, and ResNets. We vary the number of layers and width of networks to test the generalizability of results. See details in Appendix A.1.

**Tasks** We use three image classification tasks to study the tunnel effect: CIFAR-10, CIFAR-100, and CINIC-10. The datasets vary in the number of classes: 10 for CIFAR-10 and CINIC-10 and 100 for CIFAR-100, and the number of samples: 50000 for CIFAR-10 and CIFAR-100 and 250000 for CINIC-10). See details in Appendix A.2.

---

*We note that 'sufficiently large' covers most modern neural architectures, which tend to be heavily overparameterized.

We probe the effects using: *the average accuracy of linear probing, spectral analysis of representations, and the CKA similarity between representations.* Unless stated otherwise, we report the average of 3 runs.

**Accuracy of linear probing:** a linear classification layer is attached to a given layer $\ell$ of the neural network. We train this layer on the classification task and report the average accuracy. This metric measures to what extent $\ell$'s representations are linearly separable.

**Numerical rank of representations:** we compute singular values of the sample covariance matrix for a given layer $\ell$ of the neural network. Using the spectrum, we estimate the numerical rank of the given representations matrix as the number of singular values above a certain threshold $\sigma$. The threshold $\sigma$ is set to $\sigma_1 * 1e - 3$, where $\sigma_1$ is the highest singular value of the particular matrix. The numerical rank of the representations matrix can be interpreted as the measure of the degeneracy of the matrix.

**CKA similarity:** is a metric computing similarity between two representations matrices. Using this normalized index, we can identify the blocks of similar representations within the network. The definition and more details can be found in Appendix E.

**Inter and Intra class variance:** inter-class variance refers to the measure of dispersion or dissimilarity between different classes or groups in a dataset, indicating how distinct they are from each other. Intra-class variance, on the other hand, measures the variability within a single class or group, reflecting the homogeneity or similarity of data points within that class. The exact formula for computing these values can be found in Appendix F

## 2.2 The main result

Table 1 presents our main result. Namely, we report the network layer at which the tunnel begins which we define as the point at which the network reaches $95\%$ (or $98\%$) of its final accuracy. We found that all tested architectures exhibit the extractor-tunnel structure across all datasets used in the evaluation, but the relative length of the tunnel varies between architectures.

| Architecture | # layers | Dataset | $> 0.95$ | $> 0.98$ |
|:---:|:---:|:---:|:---:|:---:|
| MLP | 13 | CIFAR-10 | 4 (31%) | 5 (38%) |
| VGG | 19 | CIFAR-10 | 7 (36%) | 7 (36%) |
|  |  | CIFAR-100 | 8 (42%) | 8 (42%) |
|  |  | CINIC-10 | 7 (36%) | 7 (36%) |
| ResNet | 34 | CIFAR-10 | 20 (58%) | 29 (85%) |
|  |  | CIFAR-100 | 29 (85%) | 30 (88%) |
|  |  | CINIC-10 | 17 (50%) | 17 (50%) |

Table 1: The tunnel of various lengths is present in all tested configurations. For each architecture and dataset, we report the layer for which the *average linear probing accuracy is above* $0.95$ *and* $0.98$ *of the final performance.* The values in the brackets describe the part of the network utilized for building representations with the extractor.

We now discuss the tunnel effect using MLP-12, VGG-19, and ResNet-34 on CIFAR-10 as an example. The remaining experiments (for other architectures, datasets combinations) are available in Appendix B. As shown in Figure 1 and Figure 2, the early layers of the networks, around five for MLP and eight for VGG, are responsible for building linearly-separable representations. Linear probes attached to these layers achieve most of the network's final performance. These layers mark the transition between the extractor and the tunnel part (shaded area). In the case of ResNets, the transition takes place in deeper stages of the network at the $19^{th}$ layer.

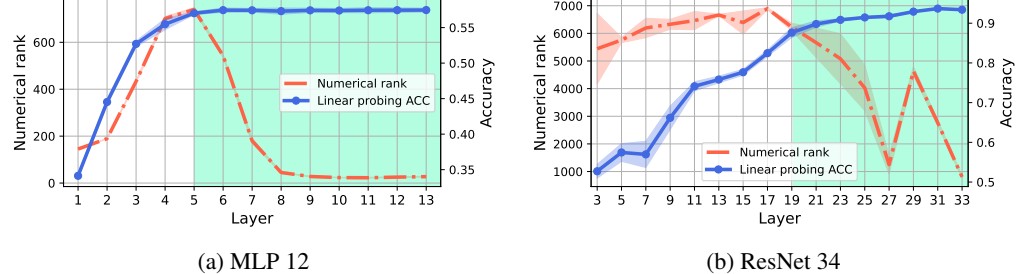

| (a) MLP 12 | (b) ResNet 34 |

Figure 2: The tunnel effect for networks trained on CIFAR-10. The blue line depicts the linear probing accuracy, and the shaded area depicts the tunnel. The red dashed line is the numerical rank of representations. The spike in the ResNet-34 representations rank coincides with the end of the penultimate residual stage.

While the linear probe performance nearly saturates in the tunnel part, the representations are further refined. Figure 2 shows that the numerical rank of the representations (red dashed line) is reduced to approximately the number of CIFAR-10 classes, which is similar to the neural collapse phenomenon observed in [10]. For ResNets, the numerical rank is more dynamic, exhibiting a spike at $29^{th}$ layer, which coincides with the end of the penultimate residual block. Additionally, the rank is higher than in the case of MLPs and VGGs.

Figure 3 reveals that for VGG-19 the inter-class representations variation decreases throughout the tunnel, meaning that representations clusters contract towards their centers. At the same time, the average distance between the centers of the clusters grows (inter-class variance). This view aligns with the observation from Figure 2, where the rank of the representations drops to values close to the number of classes. Figure 3 (right) presents an intuitive explanation of the behavior with UMAP [15] plots of the representations before and after the tunnel.

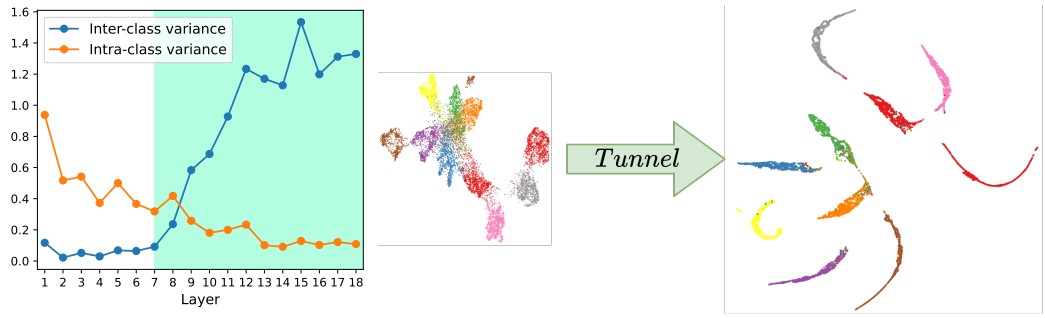

Figure 3: The tunnel compresses the representations discarding indiscriminative features. **Left:** The evolution and inter and intra-class variance of representations within the VGG-19 network. **Right:** UMAP plot of representations before ($7^{th}$ layer) and after ($18^{th}$ layer) the tunnel.

To complement this analysis, we studied the similarity of MLPs representations using the CKA index and the L1 norm of representations differences between the layers. Figure 4 shows that the representations change significantly in early layers and remain similar in the tunnel part when measured with the CKA index (left). The L1 norm of representations differences between the layers is computed on the right side of Figure 4.

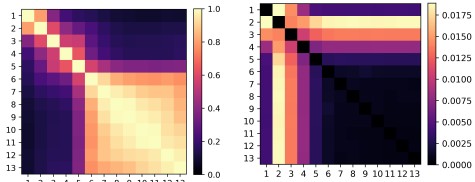

Figure 4: Representations within the tunnel are similar to each other for MLP with 12 hidden layers trained on CIFAR-10. Comparison of representations with CKA index (left) and average L1 norm of representations differences.

# 3 Tunnel effect analysis

This section provides empirical evidence contributing to our understanding of the tunnel effect. We hope that these observations will eventually lead to explanations of this phenomenon. In particular, we show that a) the tunnel develops early during training time, b) it compresses the representations and hinders OOD generalization, and c) its size is correlated with network capacity and dataset complexity.

## 3.1 Tunnel development

**Motivation** In this section, we investigate tunnel development during training. Specifically, we try to understand whether the tunnel is a phenomenon exclusively related to the representations and which part of the training is crucial for tunnel f

**Experiments** We train a VGG-19 on CIFAR-10 and save intermediate checkpoints every 10 epochs of training. We use these checkpoints to compute the layer-wise weight change during training ( Figure 5) and the evolution of numerical rank throughout the training (Figure 6).

**Results** Figure 5 shows that the split between the extractor and the tunnel is also visible in the parameters space. It could be perceived already at the early stages, and after that, its length stays roughly constant. Tunnel layers change significantly less than layers from the extractor. This result raises the question of whether the weight change affects the network's final output. Inspired by [16], we reset the weights of these layers to the state before optimization. However, the performance of the model deteriorated significantly. This suggests that although the change within the tunnel's parameters is relatively small, it plays an important role in the model's performance. Figure 6 shows that this apparent paradox can be better understood by looking at the evolution of representations' numerical rank during the very first gradient updates of the model. Throughout these steps, the rank collapses to values near-the-number of classes. It stays in this regime until the end of the training, meaning that the representations of the model evolve within a low-dimensional subspace. It remains to be understood if (and why) low-rank representations and changing weights coincide with forming linearly-separable representations.

**Takeaway** Tunnel formation is observable in the representation and parameter space.

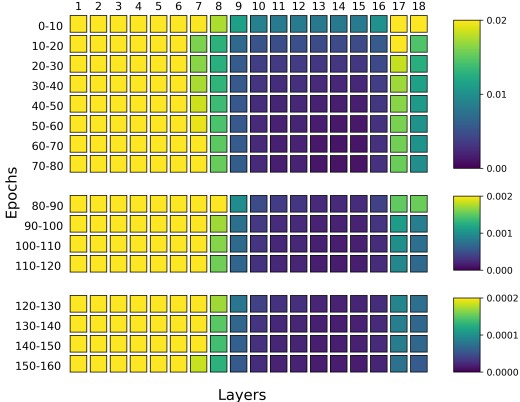

Figure 5: Early in training, tunnel layers stabilize. Color-coded: weight difference norm between consecutive checkpoints for each layer. Norm calculated as $\frac{1}{\sqrt{nm}} \|\theta_d^{\tau_1} - \theta_d^{\tau_2}\|_2$, where $\theta_d^{\tau} \in \mathbb{R}^{nm}$ is flattened weight matrix at layer $d$, checkpoint $\tau$. Values capped at 0.02 for clarity. The learning rate decayed (by $10^{-1}$) at epochs 80 and 120, and the scale adapted accordingly. Experiment: VGG-19 on CIFAR-10.

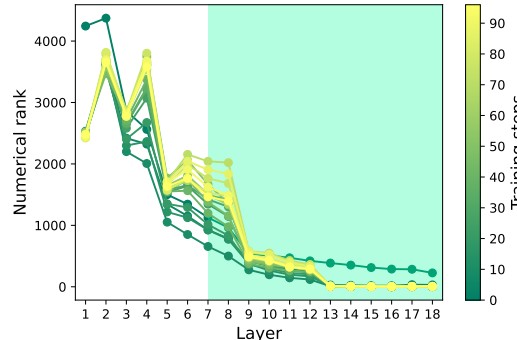

Figure 6: The representations rank for deeper layers collapse early in training. The curves present the evolution of representations' numerical rank over the first 75 training steps for all layers of the VGG-19 trained on CIFAR-10. We present a more detailed tunnel development analysis in Appendix G.

It emerges early in training and persists throughout the whole optimization. The collapse in the numerical rank of deeper layers suggest that they preserve only the necessary information required for the task.

## 3.2 Compression and out-of-distribution generalization

**Motivation** Practitioners observe intermediate layers to perform better than the penultimate ones for transfer learning [17–19]. However, the reason behind their effectiveness remains unclear [20]. In this section, we investigate whether the tunnel and, specifically, the collapse of numerical rank within the tunnel impacts the performance on out-of-distribution (OOD) data.

**Experiments** We train neural networks (MLPs, VGG-19, ResNet-34) on a source task (CIFAR-10) and evaluate it with linear probes on the OOD task, in this case, a subset of 10 classes from CIFAR-100. We report the accuracy of linear probing and the numerical rank of the representations.

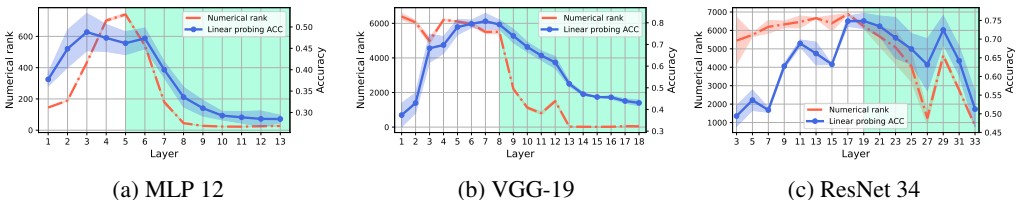

| (a) MLP 12 | (b) VGG-19 | (c) ResNet 34 |

Figure 7: The tunnel degrades the out-of-distribution performance correlated with the representations' numerical rank. The accuracy of linear probes (blue) was trained on the out-of-distribution data subset of 10 classes from CIFAR-100. The backbone was trained on CIFAR-10. The shaded area depicts the tunnel, and the red dashed line depicts the numerical rank of representations.

**Results** Our results presented in Figure 7 reveal that *the tunnel is responsible for the degradation of out-of-distribution performance*. In most of our experiments, the last layer before the tunnel is the optimal choice for training a linear classifier on external data. Interestingly, we find that the OOD performance is tightly coupled with the numerical rank of the representations, which significantly decreases throughout the tunnel.

To assess the generalization of our findings we extend the proposed experimentation setup to additional dataset. To that end, we train a model on different subsets of CIFAR-100 while evaluating it with linear probes on CIFAR-10. The results presented in Figure 8 are consistent with our initial findings. We include detailed analysis with reverse experiment (CIFAR-10 → CIFAR-100), additional architectures and datasets in the Appendix C.

In all tested scenarios, we observe a consistent relationship between the start of the tunnel and the drop in OOD performance. An increasing number of classes in the source task result in a shorter tunnel and a later drop in OOD performance. In the fixed source task experiment (Appendix C), the drop in performance occurs around the $7^{th}$ layer of the network for all tested target tasks, which matches the start of the tunnel. This observation aligns with our earlier

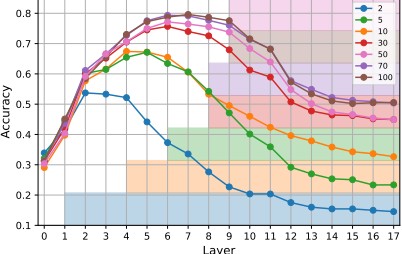

Figure 8: Fewer classes in the source task create a longer tunnel, resulting in worse OOD performance. The network is trained on subsets of CIFAR-100 with different classes, and linear probes are trained on CIFAR-10. Shaded areas depict respective tunnels .

findings suggesting that the tunnel is a prevalent characteristic of the model rather than an artifact of a particular training or dataset setup.

Moreover, we connect the coupling of the numerical rank of the representations with OOD performance, to a potential tension between the objective of supervised learning and the generalization of OOD setup. Analogous tension was observed in [21] where adversarial robustness is at odds with model's accuracy. The results in Figure 7 align with the findings presented in Figure 3, demonstrating how the tunnel compresses clusters of class-wise representations. In work [22], the authors show that reducing the variation within each class leads to lower model transferability. Our experiments support this observation and identify the tunnel as the primary contributor to this effect.

**Takeaway** Compression of representations happening in the tunnel severely degrades the OOD performance of the model which is tightly coupled with the drop of representations rank.

## 3.3 Network capacity and dataset complexity

**Motivation** In this section, we explore what factors contribute to the tunnel's emergence. Based on the results from the previous section we explore the impact of dataset complexity, network's depth, and width on tunnel emergence.

**Experiments** First, we examine the impact of networks' depth and width on the tunnel using MLPs (Figure 9), VGGs, and ResNets (Table 2) trained on CIFAR-10. Next, we train VGG-19 and ResNet34 on CIFAR-{10,100} and CINIC-10 dataset investigating the role of dataset complexity on the tunnel.

**Results** Figure 9 shows that the depth of the MLP network has no impact on the length of the extractor part. Therefore increasing the network's depth contributes only to the tunnel's length. Both extractor section and numerical rank remain relatively consistent regardless of the network's depth, starting the tunnel at the same layer. This finding suggests that overparameterized neural networks allocate a fixed capacity for a given task independent of the overall capacity of the model.

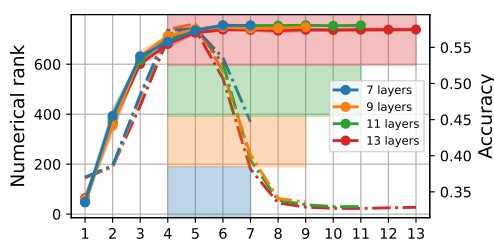

Figure 9: Networks allocate a fixed capacity for the task, leading to longer tunnels in deeper networks. The extractor is consistent across all scenarios, with the tunnel commencing at the 4th layer.

|  | $^1/_4$ | 1 | 2 |
|---|---|---|---|
| VGG-16 | 8 (50%) | 7 (44%) | 7 (44%) |
| VGG-19 | 8 (42%) | 7 (37%) | 7 (37%) |
| ResNet18 | 15 (83%) | 13 (72%) | 13 (72%) |
| ResNet34 | 24 (68%) | 20 (59%) | 24 (68%) |

Table 2: Widening networks layers results in a longer tunnel and shorter extractor. Column headings describe the factor in which we scale each model's base number of channels. The models were trained on the CIFAR-10 to the full convergence. We use the 95% threshold of probing accuracy to estimate the tunnel beginning.

Results in Table 2 indicate that the tunnel length increases as the width of the network grows, implying that representations are formed using fewer layers. However, this trend does not hold for ResNet34, as the longest tunnel is observed with the base width of the network. In the case of VGGs, the number of layers in the network does not affect the number of layers required to form representations. This aligns with the results in Figure 9.

The results presented above were obtained from a dataset with a consistent level of complexity. The data in Table 3 demonstrates that the number of classes in the dataset directly affects the length of the tunnel. Specifically, even though the CINIC-10 training dataset is three times larger than CIFAR-10, the tunnel length remains the same for both datasets. This suggests that the number of samples in the dataset does not impact the length of the tunnel. In contrast, when examining CIFAR-100 subsets, the tunnel length for both VGGs and ResNets increase. This indicates a clear relationship between the dataset's number of classes and the tunnel's length.

| model | dataset | 30% | 50% | 100% |
|---|---|---|---|---|
|  | CIFAR-10 | 6 (32%) | 7 (37%) | 7 (37%) |
| VGG-19 | CIFAR-100 | 8 (42%) | 8 (42%) | 9 (47%) |
|  | CINIC-10 | 6 (32%) | 7 (37%) | 7 (37%) |
|  | CIFAR-10 | 19 (56%) | 19 (56%) | 21 (61%) |
| ResNet34 | CIFAR-100 | 30 (88%) | 30 (88%) | 31 (91%) |
|  | CINIC-10 | 9 (27%) | 9 (27%) | 17 (50%) |

Table 3: Networks trained on tasks with fewer classes utilize fewer resources for building representations and exhibit longer tunnels. Column headings describe the size of the class subset used in training. Within the (architecture, dataset) pair, the number of gradient steps during training in all cases was the same. We use the 95% threshold of probing accuracy to estimate the tunnel beginning.

**Takeaway** Deeper or wider networks result in longer tunnels. Networks trained on datasets with fewer classes have longer tunnels.

# 4 The tunnel effect under data distribution shift

Based on the findings from the previous section and the tunnel's negative impact on transfer learning, we investigate the dynamics of the tunnel in continual learning scenarios, where large models are often used on smaller tasks typically containing only a few classes. We focus on understanding the impact of the tunnel effect on transfer learning and catastrophic forgetting [23]. Specifically, we examine how the tunnel and extractor are altered after training on a new task.

## 4.1 Exploring the effects of task incremental learning on extractor and tunnel

**Motivation** In this section, we aim to understand the tunnel and extractor dynamics in continual learning. Specifically, we examine whether the extractor and the tunnel are equally prone to catastrophic forgetting.

**Experiments** We train a VGG-19 on two tasks from CIFAR-10. Each task consists of 5 classes from the dataset. We subsequently train on the first and second tasks and save the corresponding extractors $E_t$ and tunnels $T_t$, where $t \in \{1, 2\}$ is the task number. We also save a separate classifying head for trained on each task, that we use during evaluation.

**Results** As presented in Table 4, in any combination changing $T_1$ to $T_2$ or vice versa have a marginal impact on the performance. This is quite remarkable, and suggests that the tunnel is not specific to the training task. It seems that it *compresses the representations in a task-agnostic way*. The extractor part, on the other hand, is *task-specific* and prone to forgetting as visible in the first four rows of Table 4. In the last two rows, we present two experiments that investigate how the existence of a tunnel affects the possibility of recovering from this catastrophic forgetting. In the first one, referred to as $(E_2 + T_1(FT))$, we use original data from Task 1 to retrain a classifying head attached on top of extractor $E_2$ and the tunnel $T_1$. As visible, it has minimal effect on the accuracy of the first

|                | First Task | Second Task |
|----------------|------------|-------------|
| $E_1 + T_1$    | 92.04%     | 56.8%       |
| $E_1 + T_2$    | 92.5%      | 58.04 %     |
| $E_2 + T_2$    | 50.84 %    | 93.94 %     |
| $E_2 + T_1$    | 50.66 %    | 93.72 %     |
| $E_2 + T_1(FT)$| 56.1%      | –           |
| $E_2(FT)$      | 74.4%      | –           |

Table 4: The tunnel part is task-agnostic and can be freely mixed with different extractors retaining the original performance. We test the model's performance on the first or second task using a combination of extractor $E_t$ and tunnel $T_t$ from tasks $t \in \{1, 2\}$. The last two rows $(FT)$ show how much performance can be recovered by retraining the linear probe attached to the penultimate layer $E_1 + T_1$ or the last layer of the $E_2$.

task. In the second experiment, we attach a linear probe directly to the extractor representations $(E_2(FT))$. This difference hints at a detrimental effect of the tunnel on representations' usability in continual learning.

In Appendix D.1 we study this effect further by training a tunnels on two tasks with a different number of classes, where $n_1 > n_2$. In this scenario, we observe that tunnel trained with more classes $(T_1)$ maintains the performance on both tasks, contrary to the tunnel $(T_2)$ that performs poorly on Task 1. This is in line with our previous observations in Section 2.2, that the tunnel compresses to the effective number of classes.

These results present a novel perspective in the ongoing debate regarding the layers responsible for causing forgetting. However, they do not align with the observations made in the previous study [24]. In Appendix D, we delve into the origin of this discrepancy and provide a comprehensive analysis of the changes in representations with a setup introduced with this experiment and the CKA similarity.

**Takeaway** The tunnel's task-agnostic compression of representations provides immunity against catastrophic forgetting when the number of classes is equal. These findings offer fresh perspectives on studying catastrophic forgetting at specific layers, broadening the current understanding in the literature.

## 4.2 Reducing catastrophic forgetting by adjusting network depth

**Motivation** Experiments from this section verify whether it is possible to retain the performance of the original model by training a shorter version of the network. A shallower model should also exhibit less forgetting in sequential training.

**Experiments** We train VGG-19 networks with different numbers of convolutional layers. Each network is trained on two tasks from CIFAR-10. Each task consists of 5 classes from the dataset.

**Results:** The results shown in Figure 10 indicate that training shorter networks yields similar performance compared to the original model. However, performance differences become apparent when the network becomes shorter than the extractor part in the original model. This observation aligns with previous findings suggesting that the model requires a certain capacity to perform the task effectively. Additionally, the shorter models exhibit significantly less forgetting, which corroborates the conclusions drawn in previous works [25, 26] on the importance of network depth and architecture in relation to forgetting.

**Takeaway** It is possible to train shallower networks that retain the performance of the original networks and experience significantly less forgetting. However, the shorter networks need to have at least the same capacity as the extractor part of the original network.

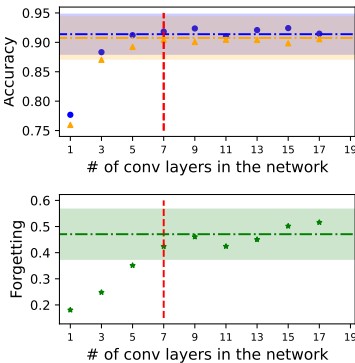

Figure 10: Training shorter networks from scratch gives a similar performance to the longer counterparts (top) and results in significantly lower forgetting (bottom). The horizontal lines denote the original model's performance. Top image: blue depicts accuracy on first task, orange depicts accuracy on the second task.

# 5 Limitations and future work

This paper empirically investigates the tunnel effect, opening the door for future theoretical research on tunnel dynamics. Further exploration could involve mitigating the tunnel effect through techniques like adjusting learning rates for specific layers. One limitation of our work is its validation within a specific scenario (image classification), while further studies on unsupervised or self-supervised methods with other modalities would shed more light and verify the pertinence of the tunnel elsewhere.

In the experiments, we observed that ResNet-based networks exhibited shorter tunnels than plain MLPs or VGGs. This finding raises the question of whether the presence of skip connections plays a role in tunnel formation. In Appendix H, we take the first step toward a deeper understanding of this relationship by examining the emergence of tunnels in ResNets without skip connections.

# 6 Related work

The analysis of representations in neural network training is an established field [27–29]. Previous studies have explored training dynamics and the impact of model width [30–35], but there is still a gap in understanding training dynamics [36, 24, 27, 37]. Works have investigated different architectures' impact on continual learning [38, 26] and linear models' behavior [39–42]. Our work builds upon studies examining specific layers' role in model performance [16, 11, 31, 36, 20, 43] and sheds light on the origins of observed behaviors [10, 44–46].

Previous works have explored the role of specific layers in model performance [16, 11, 31, 36, 20, 43]. While some studies have observed a block structure in neural network representations, their analysis was limited to ResNet architectures and did not consider continual learning scenarios. In our work, we investigate a similar phenomenon, expanding the range of experiments and gaining deeper insights into its origins. On the other hand, visualization of layer representations indicates that higher layers capture intricate and meaningful features, often formed through combinations of lower-layer features [13]. This phenomenon potentially accounts for the extension of feature extractors for complex tasks. Work [47] builds a theoretical picture that stacked sequence models tend to converge to a fixed state with infinite depth and proposes a method to compute the finite equivalent of such networks. The framework of [47] encompasses previous empirical findings of [48–50]. Independently, research on pruning methods has highlighted a greater neuron count in pruned final layers than in initial layers [51], which aligns with the tunnel's existence. Furthermore, in [52, 53], authors showed that training neural networks may lead to compressing information contained in consecutive hidden layers.

Yet another work [16] offers a different perspective, where the authors distinguish between critical and robust layers, highlighting the importance of the former for model performance, while individual layers from the latter can be reset without impacting the final performance. Our analysis builds upon this finding and further categorizes these layers into the extractor and tunnel, providing insights into their origins and their effects on model performance and generalization ability.

Our findings are also related to the Neural Collapse (NC) phenomenon [10], which has gained recent attention [44–46]. Several recent works [54–56] have extended the observation of NC and explored its impact on different layers, with a notable emphasis on deeper layers. [55] establishes a link between collapsed features and transferability. In our experiments, we delve into tunnel creation, analyzing weight changes and model behavior in a continual learning scenario, revealing the task-agnostic nature of the tunnel layers.

## 7   Conclusions

This work presents new insights into the behavior of deep neural networks during training. We discover the tunnel effect, an intriguing phenomenon in modern deep networks where they split into two distinct parts - the extractor and the tunnel. The extractor part builds representations, and the tunnel part compresses these representations to a minimum rank without contributing to the model's performance. This behavior is prevalent across multiple architectures and is positively correlated with overparameterization, i.e., it can be induced by increasing the model's size or decreasing the complexity of the task.

We emphasise that our motivation for investigating this phenomenon aimed at building a coherent picture encompassing both our experiments and evidence in the literature. Specifically, we aim to understand better how the neural networks handle the representation-building process in the context of depth.

Additionally, we discuss potential sources of the tunnel and highlight the unintuitive behavior of neural networks during the initial training phase. This novel finding has significant implications for improving the performance and robustness of deep neural networks. Moreover, we demonstrate that the tunnel hinders out-of-distribution generalization and can be detrimental in continual learning settings.

Overall, our work offers new insights into the mechanisms underlying deep neural networks. Building on consequences of the tunnel effect we derive a list of recommendations for practitioners interested in applying deep neural networks to downstream tasks. In particular, focusing on the tunnel entry features is promising when dealing with distribution shift due to its strong performance with OOD data. For continual learning, regularizing the extractor should be enough, as the tunnel part exhibits task-agnostic behavior. Skipping feature replays in deeper layers or opting for a compact model without a tunnel can combat forgetting and enhance knowledge retention. For efficient inference, excluding tunnel layers during prediction substantially cuts computation time while preserving model accuracy, offering a practical solution for resource-constrained situations.

## Acknowledgments and Disclosure of Funding

We would like to thank the reviewers for their work and insightful remarks that made this paper more complete. We especially thank Kamil Deja and other colleagues for valuable feedback and discussions during the project and writing the manuscript.

This research was funded by National Science Centre, Poland grant no 2020/39/B/ST6/01511, grant no 2022/45/B/ST6/02817, grant no 2019/35/O/ST6/03464, and grant no 2022/45/N/ST6/04098. The research was funded by Warsaw University of Technology within the Excellence Initiative: Research University (IDUB) programme. The authors have applied a CC BY license to any Author Accepted Manuscript (AAM) version arising from this submission, in accordance with the grants' open access conditions.

We gratefully acknowledge Polish high-performance computing infrastructure PLGrid (HPC Centers: ACK Cyfronet AGH) for providing computer facilities and support within computational grant no. PLG/2023/016321

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

# A    Experimental setup.

## A.1    Architectures and hyperparameters

In this section, we detail the model architectures examined in the experiments and list all hyperparameters used in the experiments.

**VGG [57]** In the main text use two types of VGG networks, namely VGG-19 and VGG-16. Both architectures consist of five stages, each consisting of a combination of convolutional layers with ReLU activation and max pooling layers. The VGG-19 has 19 layers, including 16 convolutional layers and three fully connected layers. The first two fully connected layers are followed by ReLU activation. On the other hand, VGG-16 has a total of 16 layers, including 13 convolutional layers and three fully connected layers. In additional experiments, we extend our analysis by VGG-11 and VGG-16. The base number of channels in consecutive stages for VGG architectures equals 64, 128, 256, 512, and 512.

**ResNet [58]** In experiments, we utilize two variants of the ResNet family of architectures, i.e., ResNet-18 and ResNet-34. ResNet-$N$ is a five-staged network characterized by depth, with a total of $N$ layers. The initial stage consists of a single convolutional layer – with kernel size $7 \times 7$ and 64 channels and ReLU activation, followed by max pooling $2 \times 2$, which reduces the spatial dimensions. The subsequent stages are composed of residual blocks. Each residual block typically contains two convolutional layers and introduces a shortcut connection that skips one or more layers. Each convolutional layer in the residual block is followed by batch normalization and ReLU activation. The remaining four stages in ResNet-18 and ResNet-34 architectures consist of 3x3 convolutions with the following number of channels: 64, 128, 256, and 512.

**MLP [59]** An MLP (Multi-Layer Perceptron) network is a feedforward neural network architecture type. It consists of multiple layers of artificial neurons – in our experiments, we consider MLPs with 6,8,10,12 layers with ReLU activations (except last layer, which has linear activation). In our experiments, the underlying architecture has 1024 neurons per layer.

In VGGs, MLPs, and ResNets without skips, we use the $98\%$ threshold to estimate the tunnel for the plots. In the case of ResNets, we use the $95\%$ threshold. In the case of ResNets, we report the results for the 'conv2' layers. Due to computational constraints, we randomly choose a subset of 8000 features to compute the numerical rank.

**Hyperparameters** Hyperparameters used for neural network training are presented in the leftmost Table A.1. Each column shows the values of the hyperparameters corresponding to a different architecture. The presented hyperparameters are recommended for the best performance of these models on the CIFAR-10 dataset [60]. However, in experiments focused on continual learning scenario (Section 4.2), we refrain from decaying the learning rate and shorten the network's training to 30 epochs to mimic the actual settings used in continual learning settings.

Hyperparameters used for training linear probes in our experiment are presented in the rightmost table.    Linear probes were trained with Adam optimizer instead of SGD.

| Parameter | VGG | ResNet | MLP |
|---|---|---|---|
| Learning rate (LR) | 0.1 | 0.1 | 0.05 |
| SGD momentum | 0.9 | 0.9 | 0.0 |
| Weight decay | $10^{-4}$ | $10^{-4}$ | 0 |
| Number of epochs | 160 | 164 | 1000 |
| Mini-batch size | 128 | 128 | 128 |
| LR-decay-milestones | 80, 120 | 82, 123 | - |
| LR-decay-gamma | 0.1 | 0.1 | 0.0 |

| Parameter | Value |
|---|---|
| Learning rate | 0.001 |
| Weight decay | 0 |
| Number of epochs | 30 |
| Mini-batch size | 512 |

## A.2    Datasets

In this article, we present the results of experiments conducted on following datasets: **CIFAR-10 [61]** CIFAR-10 is a widely used benchmark dataset in the field of computer vision. It consists of 60,000 color images in 10 different classes, with each class containing 6,000 images. The dataset is divided into 50,000 training images and 10,000 test images. The images in CIFAR-10 have a resolution of $32 \times 32$ pixels.

**CIFAR-100 [61]** CIFAR-100 is a dataset commonly used for image classification tasks in computer vision. It contains 60,000 color images, with 100 different classes, each containing 600 images. The dataset is split into 50,000 training images and 10,000 test images. The images in CIFAR-100 have a resolution of $32 \times 32$ pixels. Unlike CIFAR-10, CIFAR-100 offers a higher level of granularity, with more fine-grained categories such as flowers, insects, household items, and various types of animals and vehicles.

**CINIC-10 [62]** CINIC-10 is a dataset that stands as a 'bridge' between CIFAR-10 and ImageNet for image classification tasks. It combines 60,000 images of CIFAR-10, and 210,000 downsampled images of ImageNet. The images in CINIC-10 have a resolution of $32 \times 32$ pixels.

**Food-101 [63]** The Food-101 dataset is a collection of food images commonly used for image classification tasks. It contains 101 categories of food, with each category consisting of 1,000 images. The dataset covers a wide range of food items from various cuisines, including fruits, vegetables, desserts, and main dishes.

**102-Flower [64]** The 102-Flower dataset is a collection of images representing 102 different categories of flowers. Each image in the dataset has a fixed resolution of 256 pixels in both width and height. The dataset provides a diverse set of flower images.

**The Oxford-IIIT Pet Dataset [65]** The Oxford-IIIT Pet Dataset is a collection of images of cats and dogs belonging to 37 different breeds. The dataset includes a total of 7,349 images.

**Places-365 [66]** The Places-365 dataset is a large-scale dataset consisting of 365 different scene categories. It contains over 1.8 million images, each depicting a specific scene or environment. The images in the dataset have 256x256 pixels.

**STL-10 [67]** STL-10 dataset is a benchmark image dataset consisting of 10 different classes, including various animals, vehicles, and household objects. It contains a total of 5,000 training images and 8,000 test images, each with a resolution of 96 pixels by 96 pixels. The STL-10 dataset is derived from the larger ImageNet dataset but is specifically designed for low-resolution image classification tasks.

**SVHN [68]** The SVHN (Street View House Numbers) dataset is a large-scale dataset for digit recognition from real-world images. It consists of labeled images of house numbers captured from Google Street View. The dataset includes over 600,000 images for training and 26,032 images for testing. Each image is RGB and has a resolution of 32 pixels by 32 pixels.

We preprocess all datasets with standardization, additionally we rescale each image to $32px \times 32px$.

### A.3 Compute

We conducted approximately 300 experiments to finalize our work, each taking about three wall-clock hours on a single NVIDIA A5000 GPU. We had access to a server with eight NVIDIA A5000 GPUs, enabling us to parallelize our experiments and reduce total computation time. We estimate to perform over 2000 experiments (including failed ones) during the development phase of the project.

# B  Full results

## B.1  MLPs

In this section, we present the results of the tunnel effect for MLP architectures with different depths. All models are trained on CIFAR-10, and their OOD properties are evaluated on ten randomly selected classes of CIFAR-100.

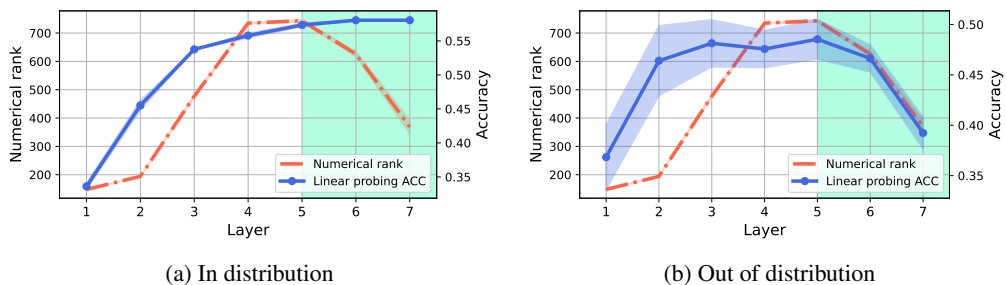



(a) In distribution     (b) Out of distribution



Figure 11: In and out of distribution linear probing performance for MLP-6 trained on CIFAR-10. The shaded area depicts the tunnel, the red dashed line depicts the numerical rank and the blue curve depicts linear probing accuracy (in and out of distribution) respectively. Out-of-distribution performance is computed with random 10 class subsets of CIFAR-100.

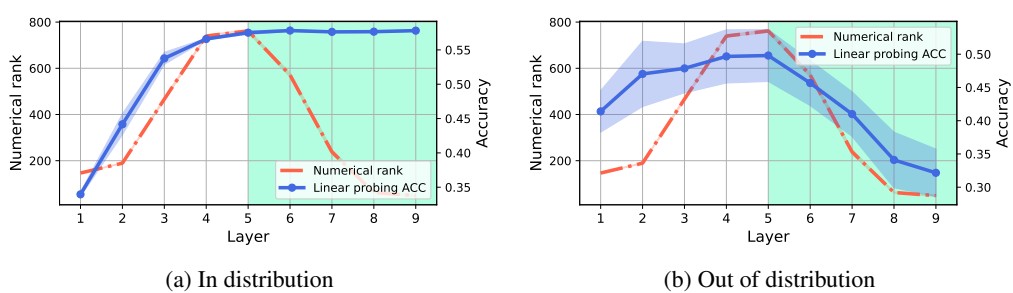



(a) In distribution     (b) Out of distribution



Figure 12: In and out of distribution linear probing performance for MLP-8 trained on CIFAR-10. The shaded area depicts the tunnel, the red dashed line depicts the numerical rank and the blue curve depicts linear probing accuracy (in and out of distribution) respectively. Out-of-distribution performance is computed with random 10 class subsets of CIFAR-100.

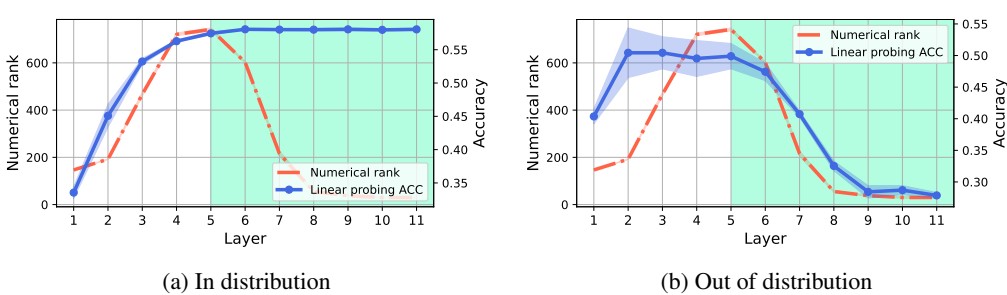



(a) In distribution     (b) Out of distribution



Figure 13: In and out of distribution linear probing performance for MLP-10 trained on CIFAR-10. The shaded area depicts the tunnel, the red dashed line depicts the numerical rank and the blue curve depicts linear probing accuracy (in and out of distribution) respectively. Out-of-distribution performance is computed with random 10 class subsets of CIFAR-100.

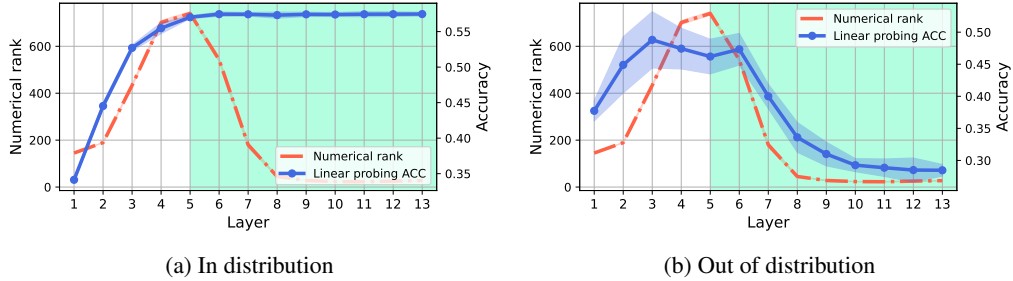

(a) In distribution                              (b) Out of distribution

Figure 14: In and out of distribution linear probing performance for MLP-12 trained on CIFAR-10. The shaded area depicts the tunnel, the red dashed line depicts the numerical rank and the blue curve depicts linear probing accuracy (in and out of distribution) respectively. Out-of-distribution performance is computed with random 10 class subsets of CIFAR-100.

## B.2 ResNet-34

In this section, we present the results of the tunnel effect for ResNet architectures with different depths. All models are trained on datasets CIFAR-10, CIFAR-100, and CINIC-10.

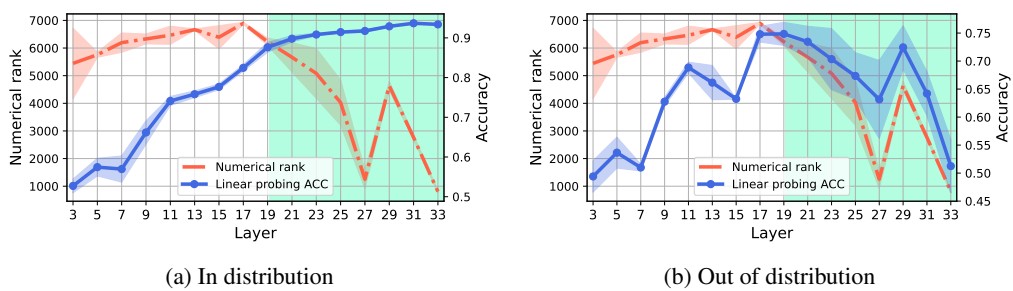

(a) In distribution                              (b) Out of distribution

Figure 15: In and out of distribution linear probing performance for ResNet-34 trained on CIFAR-10. The shaded area depicts the tunnel, the red dashed line depicts the numerical rank and the blue curve depicts linear probing accuracy (in and out of distribution) respectively. Out-of-distribution performance is computed with random 10 class subsets of CIFAR-100.

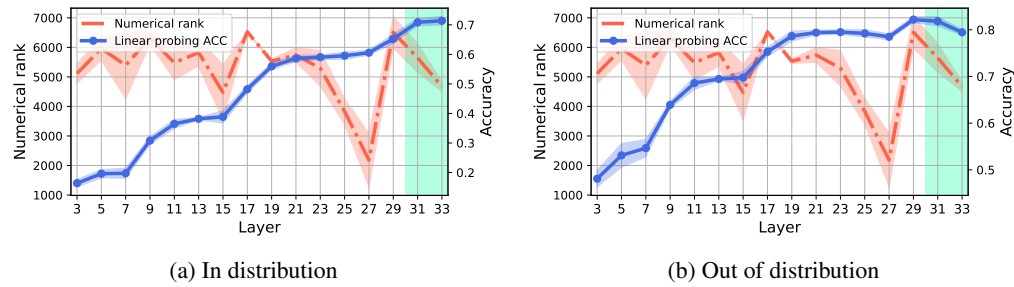

(a) In distribution                              (b) Out of distribution

Figure 16: In and out of distribution linear probing performance for ResNet-34 trained on CIFAR-100. The shaded area depicts the tunnel, the red dashed line depicts the numerical rank and the blue curve depicts linear probing accuracy (in and out of distribution) respectively. Out-of-distribution performance is computed on CIFAR-10.

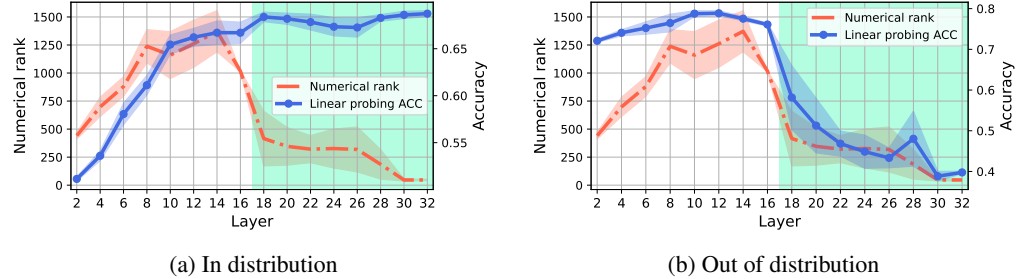

(a) In distribution    (b) Out of distribution

Figure 17: In and out of distribution linear probing performance for ResNet-34 trained on CINIC-10. The shaded area depicts the tunnel, the red dashed line depicts the numerical rank and the blue curve depicts linear probing accuracy (in and out of distribution) respectively. Out-of-distribution performance is computed on subset of ten classes from CIFAR-100.

### B.2.1 VGG-19

In this section, we present the results of the tunnel effect for VGG architectures with different depths. All models are trained on datasets CIFAR-10, CIFAR-100, and CINIC-10.

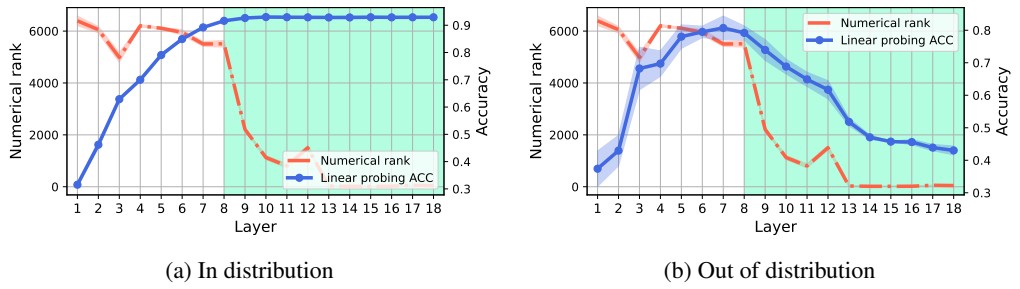

(a) In distribution    (b) Out of distribution

Figure 18: In and out of distribution linear probing performance for VGG-19 trained on CIFAR-10. The shaded area depicts the tunnel, the red dashed line depicts the numerical rank and the blue curve depicts linear probing accuracy (in and out of distribution) respectively. Out-of-distribution performance is computed with random 10 class subsets of CIFAR-100.

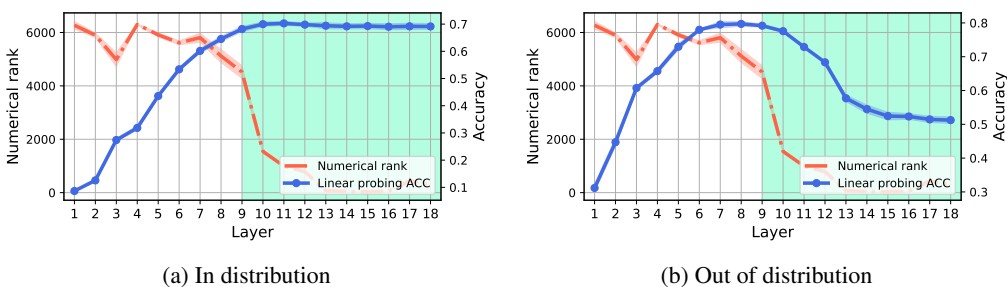

(a) In distribution    (b) Out of distribution

Figure 19: In and out of distribution linear probing performance for VGG-19 trained on CIFAR-100. The shaded area depicts the tunnel, the red dashed line depicts the numerical rank and the blue curve depicts linear probing accuracy (in and out of distribution) respectively. Out-of-distribution performance is computed on CIFAR-10.

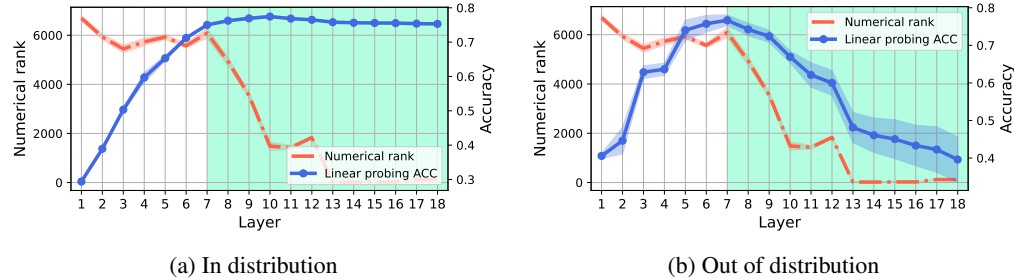

(a) In distribution          (b) Out of distribution

Figure 20: In and out of distribution linear probing performance for VGG-19 trained on CINIC-10. The shaded area depicts the tunnel, the red dashed line depicts the numerical rank and the blue curve depicts linear probing accuracy (in and out of distribution) respectively. Out-of-distribution performance is computed on subset of ten classes from CIFAR-100.

## B.3 Dataset complexity experiments

### B.3.1 ResNet-34

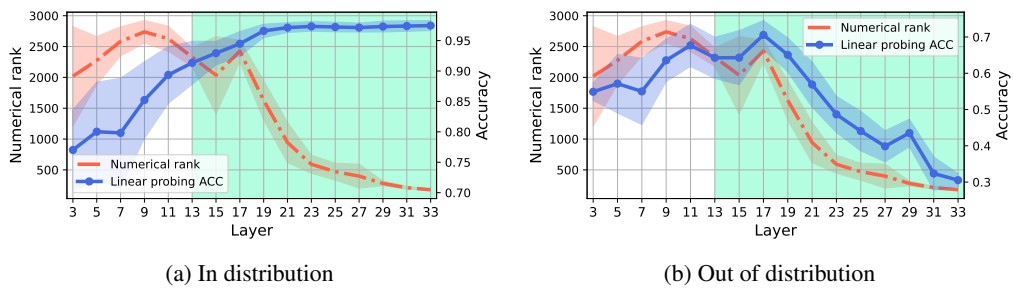

(a) In distribution

(b) Out of distribution

Figure 21: In and out of distribution linear probing performance for ResNet-34 trained on a 3-class subset of CIFAR-10. The shaded area depicts the tunnel, the red dashed line depicts the numerical rank, and the blue curve depicts linear probing accuracy (in and out of distribution) respectively. Out-of-distribution performance is computed with random 10 class subsets of CIFAR-100.

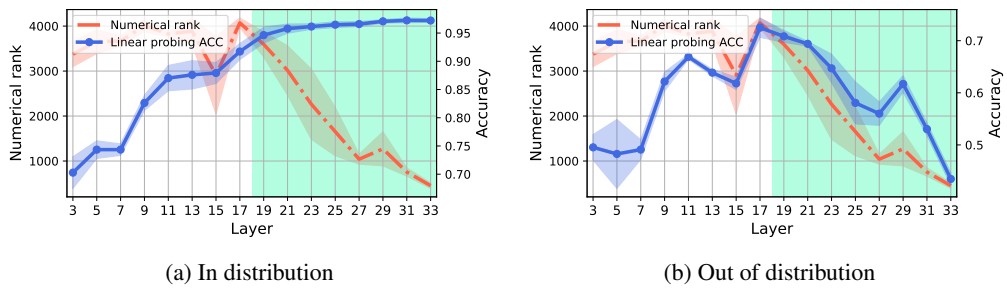

(a) In distribution

(b) Out of distribution

Figure 22: In and out of distribution linear probing performance for ResNet-34 trained on a 5-class subset of CIFAR-10. The shaded area depicts the tunnel, the red dashed line depicts the numerical rank, and the blue curve depicts linear probing accuracy (in and out of distribution) respectively. Out-of-distribution performance is computed with random 10 class subsets of CIFAR-100.

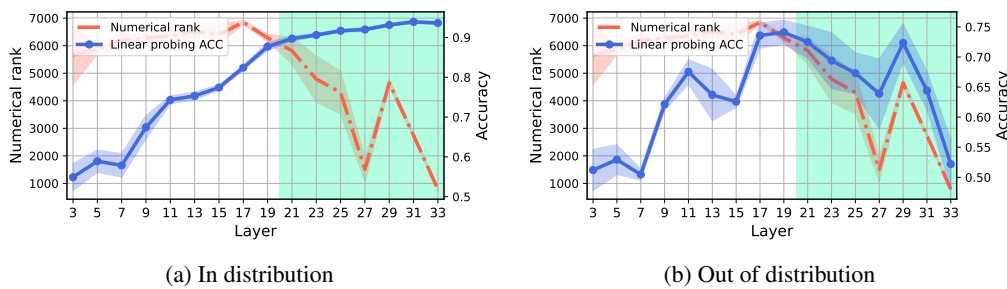

(a) In distribution

(b) Out of distribution

Figure 23: In and out of distribution linear probing performance for ResNet-34 trained on CIFAR-10. The shaded area depicts the tunnel, the red dashed line depicts the numerical rank, and the blue curve depicts linear probing accuracy (in and out of distribution) respectively. Out-of-distribution performance is computed with random 10 class subsets of CIFAR-100.

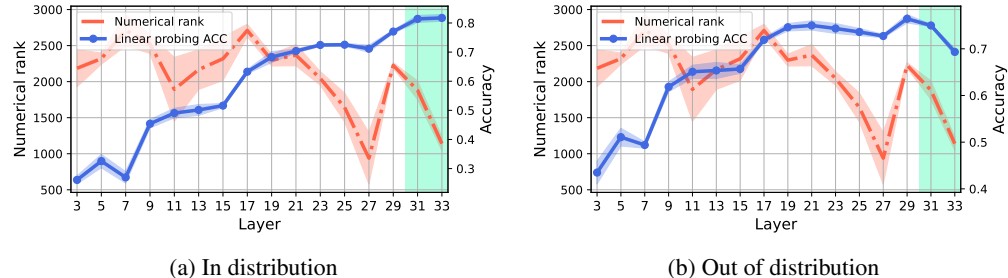

(a) In distribution

(b) Out of distribution

Figure 24: In and out of distribution linear probing performance for ResNet-34 trained on a 30-class subset of CIFAR-100. The shaded area depicts the tunnel, the red dashed line depicts the numerical rank, and the blue curve depicts linear probing accuracy (in and out of distribution) respectively. Out-of-distribution performance is computed on CIFAR-10.

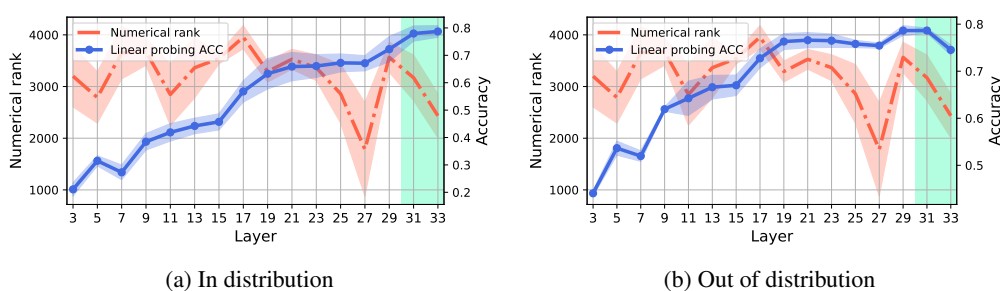

(a) In distribution

(b) Out of distribution

Figure 25: In and out of distribution linear probing performance for ResNet-34 trained on a 50-class subset of CIFAR-100. The shaded area depicts the tunnel, the red dashed line depicts the numerical rank, and the blue curve depicts linear probing accuracy (in and out of distribution) respectively. Out-of-distribution performance is computed on CIFAR-10.

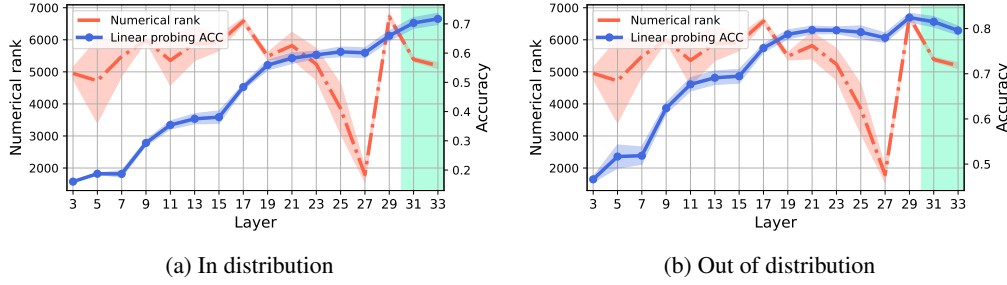

(a) In distribution

(b) Out of distribution

Figure 26: In and out of distribution linear probing performance for ResNet-34 trained on CIFAR-100. The shaded area depicts the tunnel, the red dashed line depicts the numerical rank, and the blue curve depicts linear probing accuracy (in and out of distribution) respectively. Out-of-distribution performance is computed on CIFAR-10.

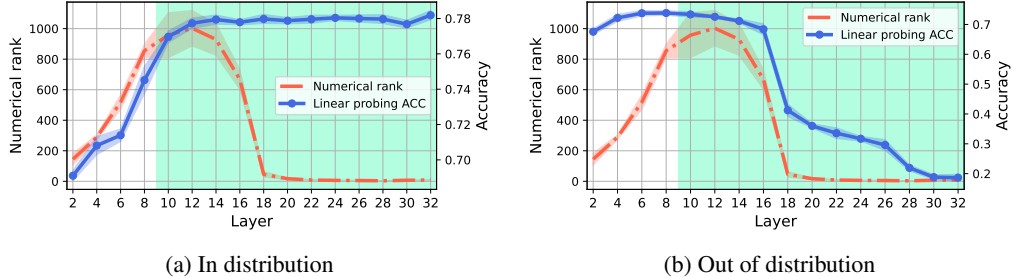

(a) In distribution        (b) Out of distribution

Figure 27: In and out of distribution linear probing performance for ResNet-34 trained on a 3-class subset of CINIC-10. The shaded area depicts the tunnel, the red dashed line depicts the numerical rank, and the blue curve depicts linear probing accuracy (in and out of distribution) respectively. Out-of-distribution performance is computed on CIFAR-10.

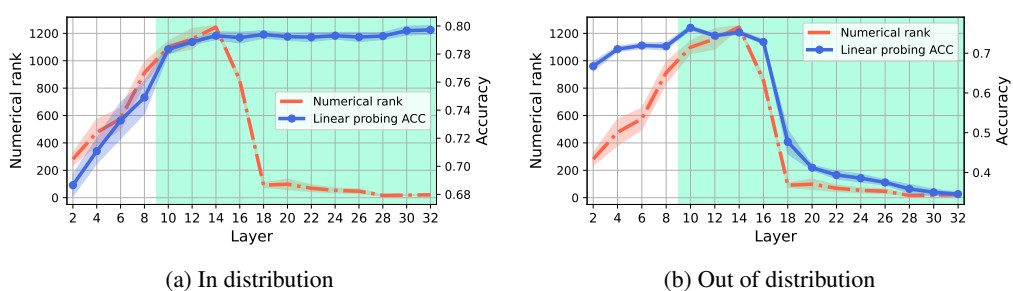

(a) In distribution        (b) Out of distribution

Figure 28: In and out of distribution linear probing performance for ResNet-34 trained on a 5-class subset of CINIC-10. The shaded area depicts the tunnel, the red dashed line depicts the numerical rank, and the blue curve depicts linear probing accuracy (in and out of distribution) respectively. Out-of-distribution performance is computed on CIFAR-10.

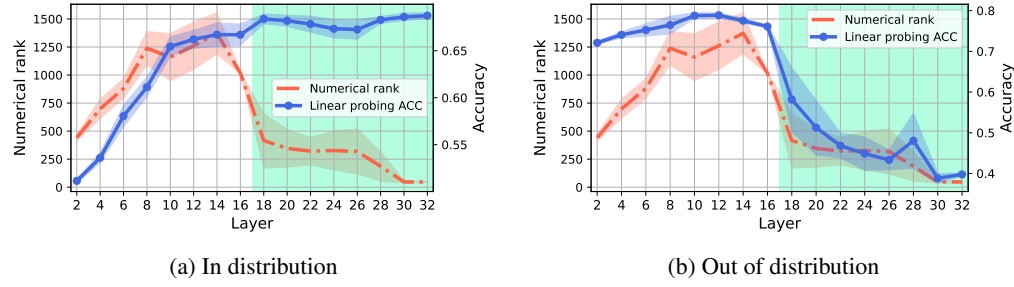

(a) In distribution        (b) Out of distribution

Figure 29: In and out of distribution linear probing performance for ResNet-34 trained on CINIC-10. The shaded area depicts the tunnel, the red dashed line depicts the numerical rank, and the blue curve depicts linear probing accuracy (in and out of distribution) respectively. Out-of-distribution performance is computed on CIFAR-10.

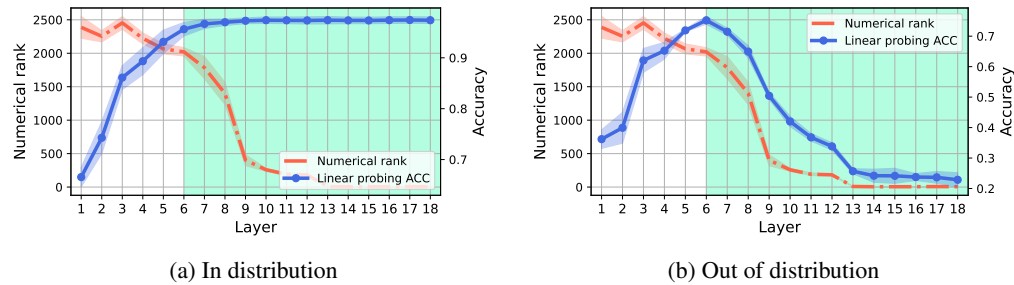

(a) In distribution          (b) Out of distribution

Figure 30: In and out of distribution linear probing performance for VGG-19 trained on a 3-class subset of CIFAR-10. The shaded area depicts the tunnel, the red dashed line depicts the numerical rank, and the blue curve depicts linear probing accuracy (in and out of distribution) respectively. Out-of-distribution performance is computed with random 10 class subsets of CIFAR-100.

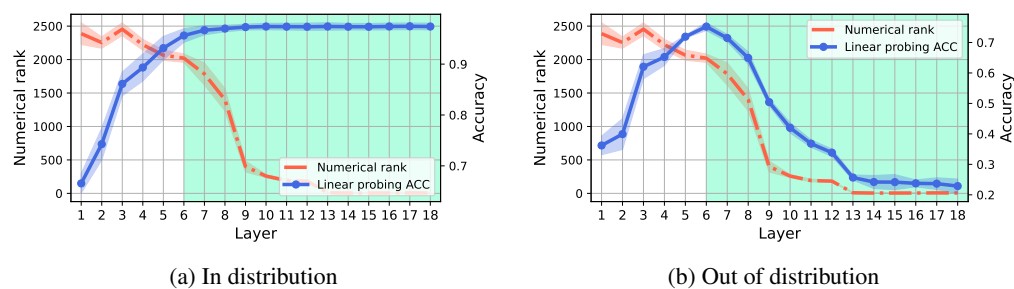

(a) In distribution          (b) Out of distribution

Figure 31: In and out of distribution linear probing performance for VGG-19 trained on a 5-class subset of CIFAR-10. The shaded area depicts the tunnel, the red dashed line depicts the numerical rank, and the blue curve depicts linear probing accuracy (in and out of distribution) respectively. Out-of-distribution performance is computed with random 10 class subsets of CIFAR-100.

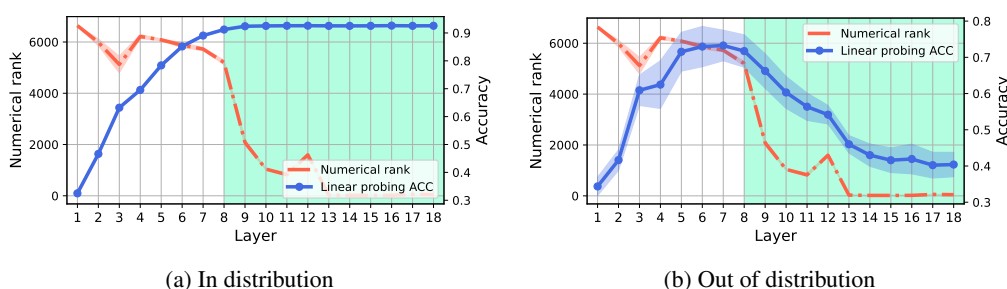

(a) In distribution          (b) Out of distribution

Figure 32: In and out of distribution linear probing performance for VGG-19 trained on CIFAR-10. The shaded area depicts the tunnel, the red dashed line depicts the numerical rank, and the blue curve depicts linear probing accuracy (in and out of distribution) respectively. Out-of-distribution performance is computed with random 10 class subsets of CIFAR-100.

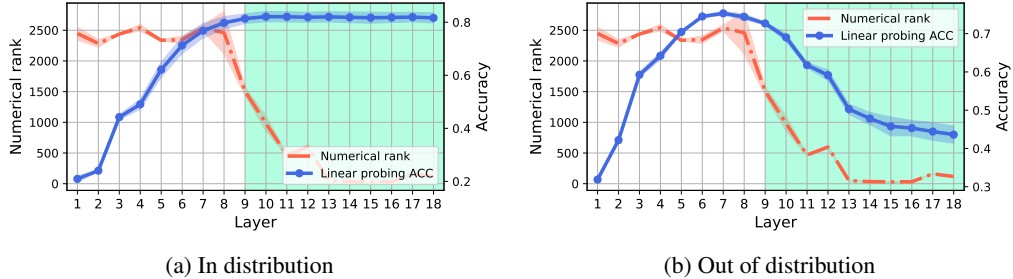

(a) In distribution        (b) Out of distribution

Figure 33: In and out of distribution linear probing performance for VGG-19 trained on a 30-class subset of CIFAR-100. The shaded area depicts the tunnel, the red dashed line depicts the numerical rank, and the blue curve depicts linear probing accuracy (in and out of distribution) respectively. Out-of-distribution performance is computed on CIFAR-10.

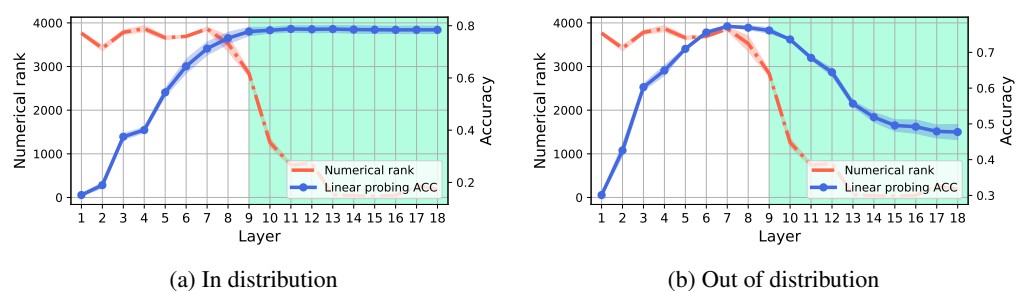

(a) In distribution        (b) Out of distribution

Figure 34: In and out of distribution linear probing performance for VGG-19 trained on a 50-class subset of CIFAR-100. The shaded area depicts the tunnel, the red dashed line depicts the numerical rank, and the blue curve depicts linear probing accuracy (in and out of distribution) respectively. Out-of-distribution performance is computed on CIFAR-10.

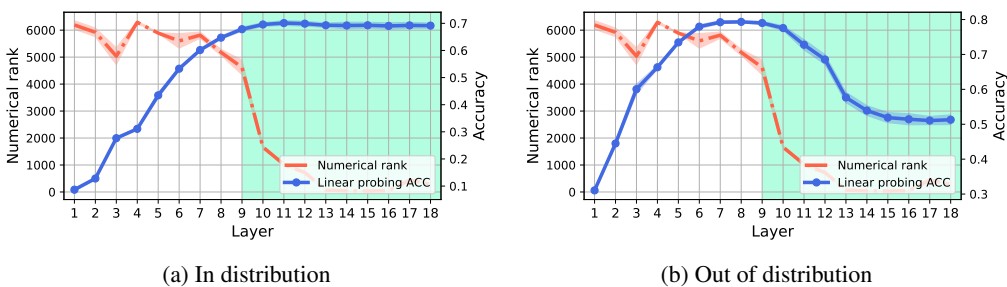

(a) In distribution        (b) Out of distribution

Figure 35: In and out of distribution linear probing performance for VGG-19 trained on CIFAR-100. The shaded area depicts the tunnel, the red dashed line depicts the numerical rank, and the blue curve depicts linear probing accuracy (in and out of distribution) respectively. Out-of-distribution performance is computed on CIFAR-10.

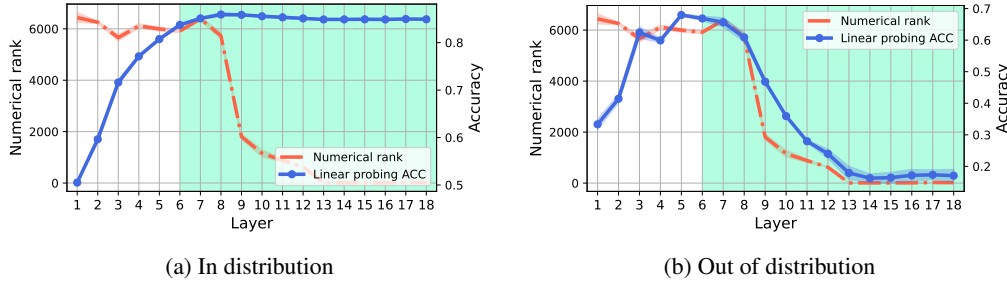

(a) In distribution

(b) Out of distribution

Figure 36: In and out of distribution linear probing performance for VGG-19 trained on a 3-class subset of CINIC-10. The shaded area depicts the tunnel, the red dashed line depicts the numerical rank, and the blue curve depicts linear probing accuracy (in and out of distribution) respectively. Out-of-distribution performance is computed on CIFAR-10.

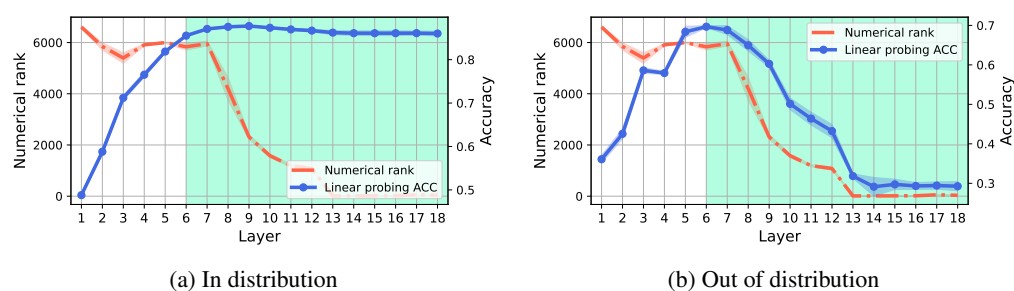

(a) In distribution

(b) Out of distribution

Figure 37: In and out of distribution linear probing performance for VGG-19 trained on a 5-class subset of CINIC-10. The shaded area depicts the tunnel, the red dashed line depicts the numerical rank, and the blue curve depicts linear probing accuracy (in and out of distribution) respectively. Out-of-distribution performance is computed on CIFAR-10.

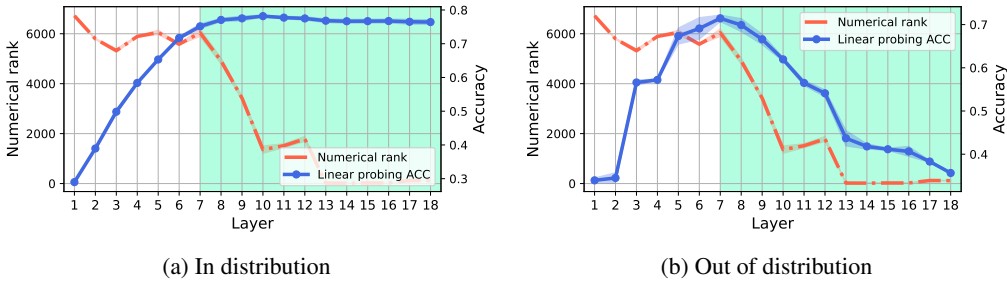

(a) In distribution

(b) Out of distribution

Figure 38: In and out of distribution linear probing performance for VGG-19 trained on CINIC-10. The shaded area depicts the tunnel, the red dashed line depicts the numerical rank, and the blue curve depicts linear probing accuracy (in and out of distribution) respectively. Out-of-distribution performance is computed on CIFAR-10.

# C Out of distribution generalization - extended results

In this experiment, we aim to determine if the tunnel consistently decreases the performance of models on out-of-distribution (OOD) datasets. To achieve this, we trained VGG-19 and ResNet34 models on CIFAR-10 and conducted linear probing on various OOD datasets. The results, depicted in Figure 39, are consistent across both the tested models and the datasets used. Notably, in all cases except for the training dataset (CIFAR-10), we observe a decline in performance starting from the beginning of the tunnel and continuing to degrade further. In the case of ResNet-34, there is a spike in performance at the $29^{th}$ layer, which aligns with the findings in the main paper. Interestingly, the dataset that exhibits the least deterioration is STL-10. This dataset consists of 10 classes, 9 of which overlap with classes found in CIFAR-10. However, the images in STL-10 are sampled from the ImageNet dataset. These results suggest that models can generalize well to OOD data that share semantic similarities with the in-distribution data. Note that the linear probing performance was normalized for better presentation.

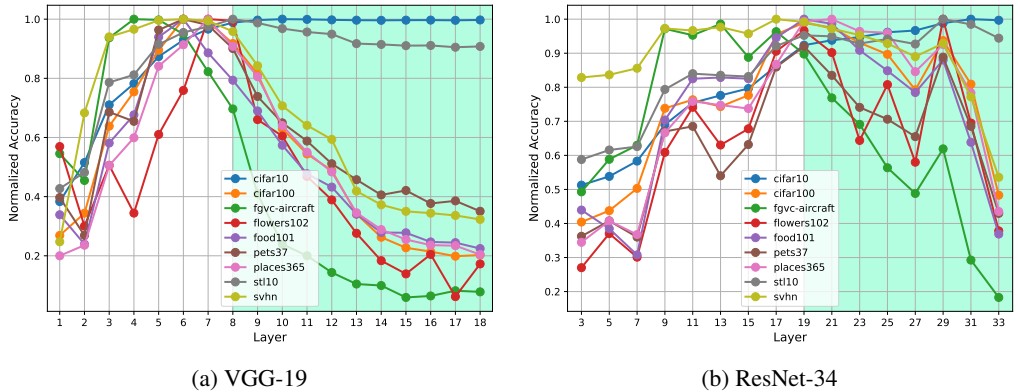

| (a) VGG-19 | (b) ResNet-34 |

Figure 39: Out of distribution normalized linear probing performance for different datasets. The shaded area depicts the tunnel, different colors depict the linear probing performance on given dataset. Note that all the results are normalized for clarity of presentation.

The following experiment complements the analysis presented in the main paper, aiming to further explore the degradation of out-of-distribution performance caused by the tunnel effect. In this particular setup, the network is trained using CIFAR-10, and linear probes are trained and evaluated using subsets of CIFAR-100 with varying numbers of classes. The results, depicted in Figure 40, consistently demonstrate that regardless of the number of classes used to train the linear probes, the tunnel effect consistently leads to a decline in their performance. These findings confirm our observations from the main paper, indicating that the tunnel effect is a prevalent characteristic of the model rather than a peculiar artifact of the dataset or training setup.

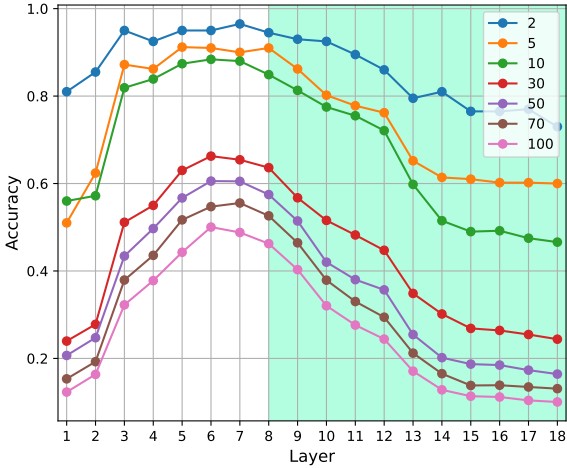

Figure 40: Source task with a fixed number of classes results in a tunnel consistently degrading the OOD performance for a different number of classes. VGG-19 is trained on CIFAR-10 and linear probes are trained on different subsets of CIFAR-100 with different numbers of classes. The tunnel is marked with a shaded color.

# D    Exploring the effects of task incremental learning on extractor and tunnel – extended results

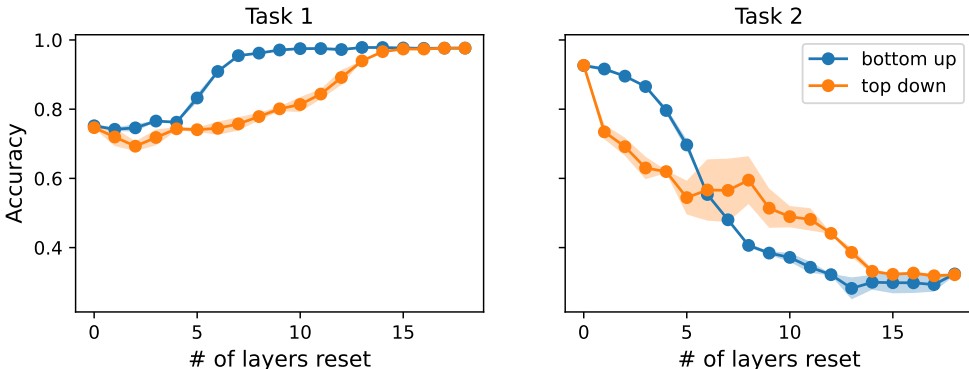

Figure 41: Substituting layer experiment. VGG-19 trained on the sequence of two tasks on split-CIFAR10. First task 3 class, second task 7 classes.

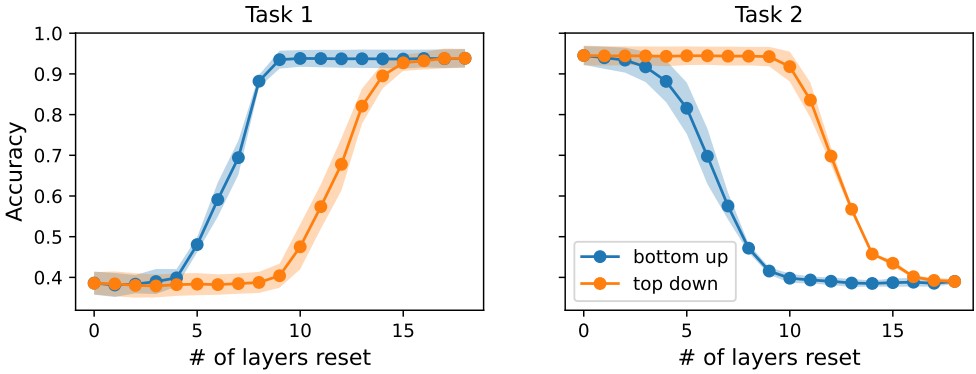

Figure 42: Substituting layer experiment. VGG-19 trained on the sequence of two tasks on split-CIFAR10. First task 5 class, second task 5 classes.

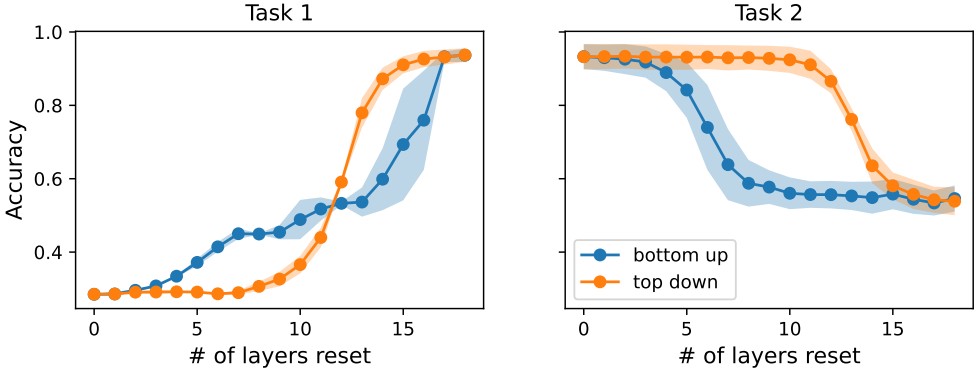

Figure 43: Substituting layer experiment. VGG-19 trained on the sequence of two tasks on split-CIFAR10. First task 7 class, second task 3 classes.

In this section we discuss in greater details experiment from section 4.1. First, we focus on examining reset layers experiment in case of sequence of tasks with different number of classes in section D.1. Next, we discuss the the discrepancies between our results and results presented in work [24].

### D.1 Different number of classes in source and target tasks.

In this experiment, our aim is to gain a better understanding of tunnel immunity to catastrophic forgetting. Specifically, we are interested in exploring scenarios where the number of classes differs in each task. To analyze this scenario, we conducted three experiments using the VGG-19 network. We trained the network on sequences of two tasks, each composed of CIFAR-10 classes with different splits: $(3, 7)$, $(5, 5)$, and $(7, 3)$.

During training, we saved the model after completing the first and second tasks, denoted as $M_1$ and $M_2$ respectively. When we refer to $M_1^{1:x} + M_2^{x+1:n}$, we mean that the network consists of the first $x$ layers with parameters from after completing the first task, combined with the remaining $n - x$ layers from the network after completing the second task.

Here, instead of a table we present the results using plots, see Figure 41 for the reference. The y-axis values represent the accuracy of the model when substituting a certain number of layers, denoted as $x$. The blue plot represents the situation where we substitute layers starting from the bottom $(M_1^{1:x} + M_2^{x+1:n})$, while the orange plot represents the opposite scenario $(M_2^{1:x} + M_1^{x+1:n})$. Please note the change in subscripts.

In Figure 42, we observe that when the tasks have an equal number of classes, the tunnel is preserved perfectly. Specifically, substituting 10 layers from the top down does not affect the performance on the second task, and substituting more than 8 layers does not yield any improvement on the first task.

Conversely, in Figure 41, substituting more than 7 layers from the bottom up does not lead to any improvement in the second task. Additionally, substituting any layers from the top down actually harms the performance on the second task. This suggests that while the network encountered more classes in the second task, it built upon the existing tunnel, maintaining its performance on the first task.

In the opposite scenario, where the second task involves fewer classes, a reverse situation is observed. Substituting any layers from the top down negatively impacts the performance on the first task, while substituting 10 layers from the top down does not affect the performance on the second task. This suggests that the network successfully reused a portion of the tunnel from the first task while discarding the unnecessary part.

### D.2 On the primary source of catastrophic forgetting on split-CIFAR10 task.

There is an ongoing discussion surrounding the layers responsible for driving the phenomenon of forgetting. In a study [24], authors claim that "Higher layers are the primary source of catastrophic forgetting on split CIFAR-10 task." However, our findings present a different perspective compared to the conclusions drawn in that research. Specifically, the results presented in Section 4.1 and Section D.1 indicate that there exist continual learning scenarios where the deeper layers do not contribute to catastrophic forgetting. Instead, we show that in certain scenarios the earlier layers are responsible for performance degradation, while the deeper layers remain unaffected due to their task-agnostic nature. This insight is of particular significance because many studies have built upon the assumption that mainly deeper layers are responsible for catastrophic forgetting, potentially leading to inadequate or inefficient continual learning mechanisms [69, 70].

It is important to note that the tunnel hypothesis effect holds for overparameterized networks. In contrast, the authors of [24] evaluated their claims using the VGG-13 network, with the width of the layers reduced by a factor of four. This discrepancy plays a crucial role in tunnel formation, as it reduces the model's capacity. Figures 44- 49 illustrate the disparity between these models in the reset experiment.

From this comparison, the main conclusion emerges that the question of "which layers are the primary source of catastrophic forgetting?" is nuanced and contingent upon multiple factors.

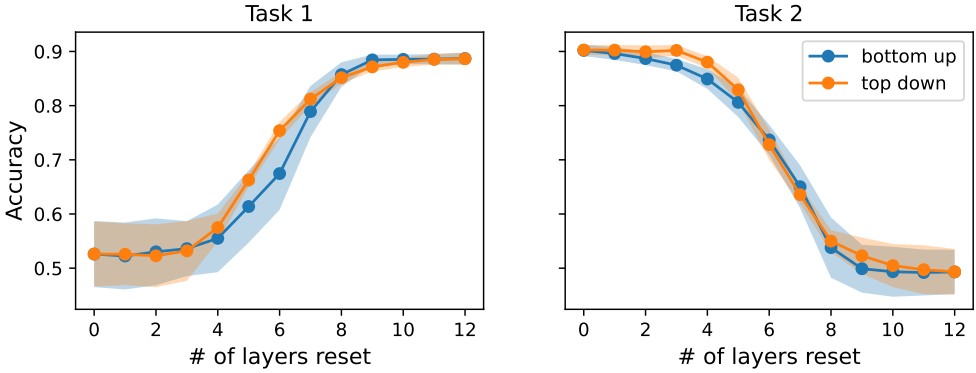

Figure 44: Substituting layer experiment. VGG-13, width factor = 0.25, trained on the sequence of two tasks on split-CIFAR10.

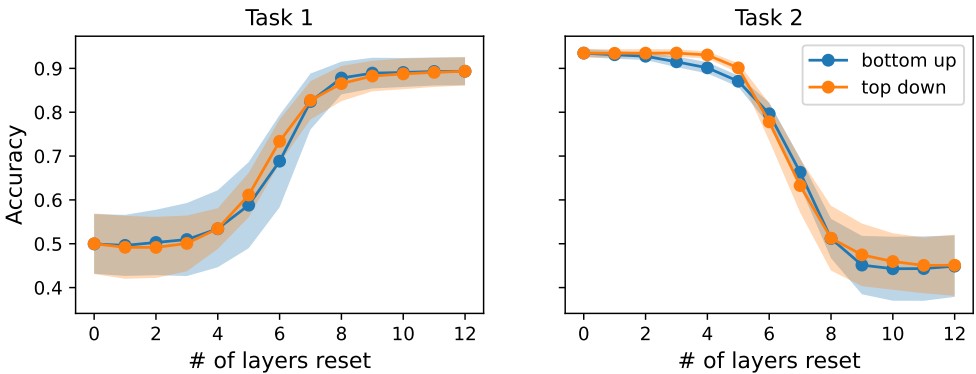

Figure 45: Substituting layer experiment. VGG-13, width factor = 1, trained on the sequence of two tasks on split-CIFAR10.

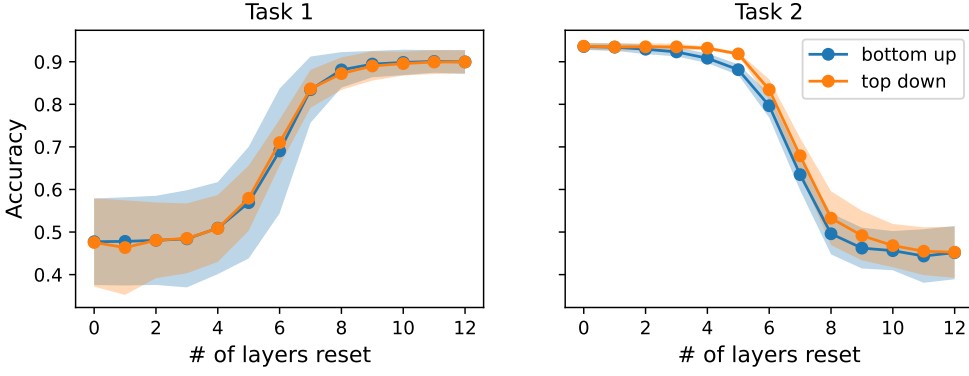

Figure 46: Substituting layer experiment. VGG-13, width factor = 2, trained on the sequence of two tasks on split-CIFAR10.

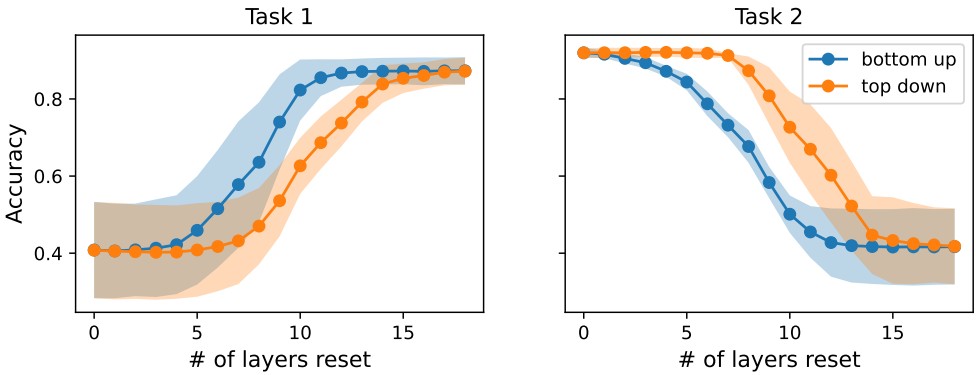

Figure 47: Substituting layer experiment. VGG-19, width factor = 0.25, trained on the sequence of two tasks on split-CIFAR10.

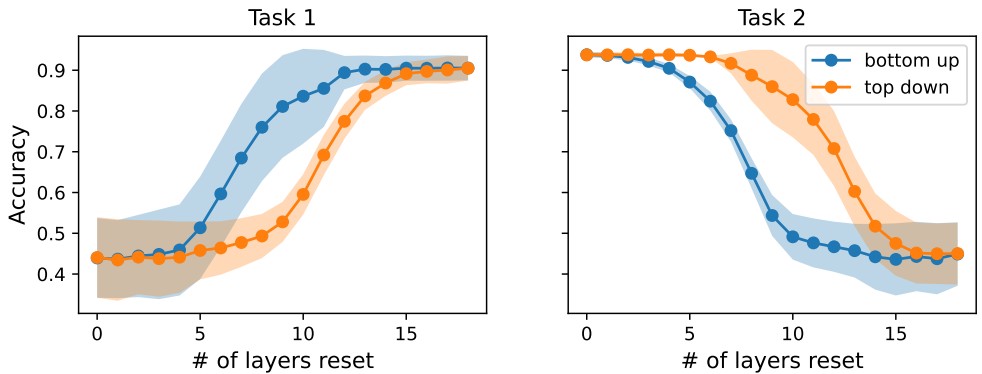

Figure 48: Substituting layer experiment. VGG-19, width factor = 1, trained on the sequence of two tasks on split-CIFAR10.

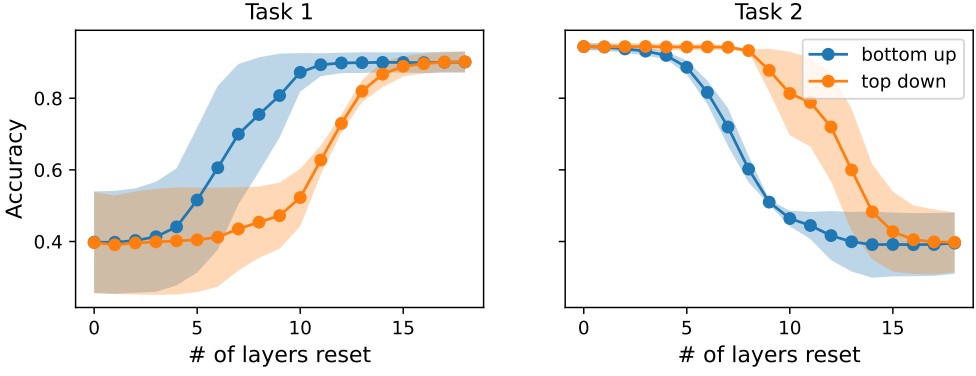

Figure 49: Substituting layer experiment. VGG-19, width factor = 2, trained on the sequence of two tasks on split-CIFAR10.

# E  CKA similarity

The Centered Kernel Alignment (CKA) similarity is a measure commonly used in machine learning and neuroscience to quantify the similarity between two representations or feature spaces. It provides a way to assess the similarity of representations learned by different models or layers, even when the representations may have different dimensionalities. CKA is invariant to orthogonal transformations, such as isotropic scaling, permutations, reflections and rotations. This invariance property is particularly valuable when comparing representations that have undergone different preprocessing or normalization steps. By accounting for the underlying relationships between representations while being insensitive to orthogonal transformations, CKA enables a more meaningful and reliable assessment of similarity, aiding in tasks such as model comparison, representation learning, and understanding the neural code. For computational reasons, we use a modification of CKA index given by the following formula.

**CKA similarity:** Let $\mathbf{X}_i \in \mathbb{R}^{m \times p_1}$, $\mathbf{Y}_i \in \mathbb{R}^{m \times p_2}$ be the representations matrices from $i^{th}$ minibatch of two layers of $m$ samples and $p_1$ and $p_2$ number of features respectively. Similarly to [11], we estimate CKA index by averaging over $k$ mini-batches:

$$sCKA = \frac{\frac{1}{k}\sum_{i=1}^{k} \text{HSIC}\left(\mathbf{X}_i\mathbf{X}_i^\top, \mathbf{Y}_i\mathbf{Y}_i^\top\right)}{\sqrt{\frac{1}{k}\sum_{i=1}^{k}\text{HSIC}\left(\mathbf{X}_i\mathbf{X}_i^\top, \mathbf{X}_i\mathbf{X}_i^\top\right)}\sqrt{\frac{1}{k}\sum_{i=1}^{k}\text{HSIC}\left(\mathbf{Y}_i\mathbf{Y}_i^\top, \mathbf{Y}_i\mathbf{Y}_i^\top\right)}}, \tag{1}$$

where HSIC is an unbiased estimate or of the HSIC score [11]:

$$\text{HSIC}(\mathbf{K}, \mathbf{L}) = \frac{1}{n(n-3)}\left(\text{tr}(\tilde{\mathbf{K}}\tilde{\mathbf{L}}) + \frac{\mathbf{1}^\top\tilde{\mathbf{K}}\mathbf{1}\mathbf{1}^\top\tilde{\mathbf{L}}\mathbf{1}}{(n-1)(n-2)} - \frac{2}{n-2}\mathbf{1}^\top\tilde{\mathbf{K}}\tilde{\mathbf{L}}\mathbf{1}\right), \tag{2}$$

where $\tilde{\mathbf{L}} = \mathbf{L} - diag(\mathbf{L})$.

CKA is a normalized similarity index, hence value 1 means that representations matrices are identical.

In Figure 50, we present the plots of representations similarity for ResNet-18 and VGG-19 networks.

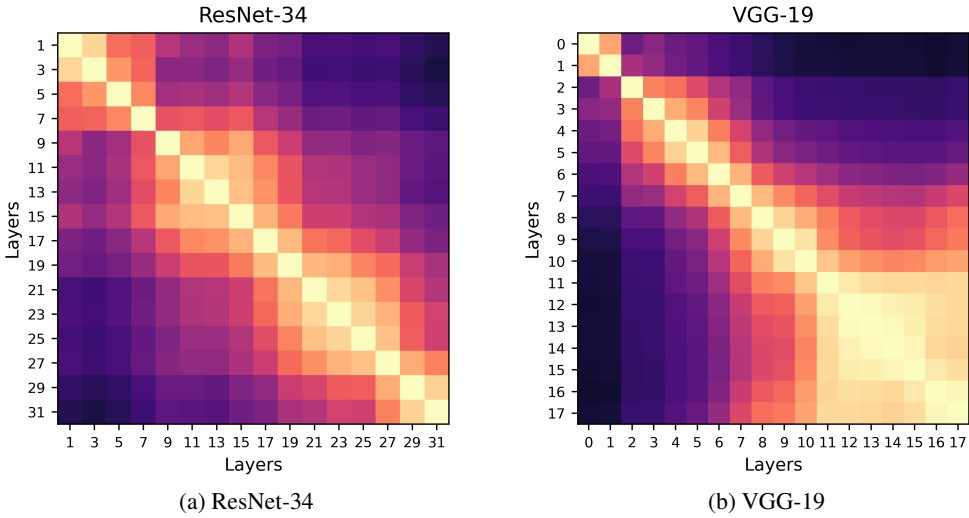

(a) ResNet-34                    (b) VGG-19

Figure 50: CKA-similarity across network's layers for ResNet-34 and VGG-19.

# F  Inter and Intra class variance

Understanding the concepts of inter-class and intra-class variance is particularly important in the context of deep neural network representations analysis for classification tasks. In this context,

inter-class variance refers to the variability between different classes or categories of data. On the one hand, they capture the representation of the linear separability of a given task [71]. On the other hand, intra-class variance is an indicator of representations transferability [72, 73].

Let $\mathbf{X}_j \in \mathbb{R}^{t_j \times p}$ be the representations matrix for samples from $j^{th}$ class. $\frac{1}{C} \sum_{j=1}^{C} \left( \frac{1}{t_j} \sum_{x_i \in \mathbf{X}_j} \|\mathbf{f}_i - \mu(\mathbf{X}_j)\|^2 \right)$ is the intra-class variance, where $\mathbf{f}_i$ is a representation of sample $x_i$, $\mu(\mathbf{X}_j)$ is the mean representation of representations matrix $\mathbf{X}_j$, and $C$ is the number of classes. Then $\frac{1}{C(C-1)} \sum_{j=1}^{C} \sum_{k=1,k\neq j}^{C} \|\mu(\mathbf{X}_j) - \mu(\mathbf{X}_k)\|^2$ is the inter-class variance.

## G Tunnel development

In this section, we provide a more detailed analysis of the evolution of numerical rank in VGG-19 dataset. In this experiment, we save the checkpoint of the network every epoch and calculate its numerical rank. The results are depicted in Figure 51.

Initially, during the early epochs, the rank collapses primarily in the deeper layers. Throughout the training process, two distinct patterns can be observed. Firstly, the numerical rank of representations from the earlier layers tends to increase. Secondly, the numerical rank of representations from the deeper layers decreases. Interestingly, the place of this transition aligns with the beginning of the tunnel in the network. Once the numerical rank in deeper layers collapsed in the first gradient steps, as shown in Figure 6, it remained collapsed throughout the whole training.

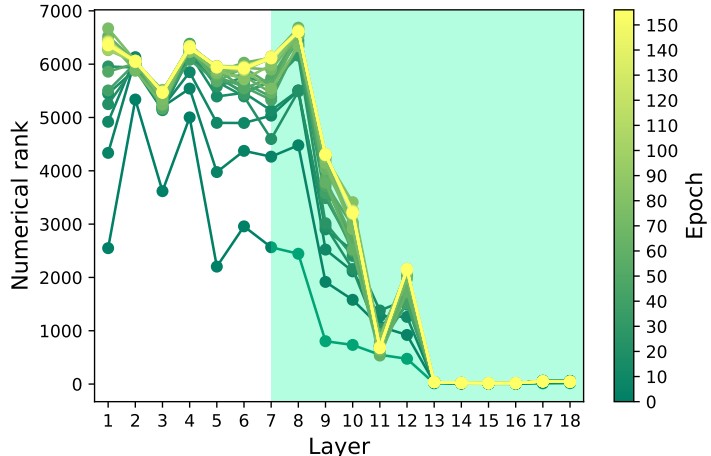

Figure 51: Evolution of numerical rank of VGG-19 representations throughout the training on CIFAR-10.

## H ResNets without skip connections

In this section, we delve into the impact of skip connections on the formation of the tunnel effect. To investigate this relationship, we trained ResNet models (ResNet-18 and ResNet-34) without skip connections on CIFAR-10 and conducted the same analysis used in the main paper. Specifically, we examined the linear probing performance for both in-distribution and out-of-distribution data and estimated the representations' numerical rank. The results, depicted in Figure 52 and Figure 53, highlight the significance of skip connections in the formation of the tunnel effect. Firstly, in the absence of skip connections (Plainnets), the tunnel effect is slightly more pronounced, with the model's performance saturating two layers earlier than standard ResNet networks. Secondly, the rank of the representations exhibits a more predictable pattern without the spike at $29^{th}$ layer. This suggests that the spike in the numerical rank and in OOD performance is related to the skip connections. Interestingly, the numerical rank in both networks is higher than in the case of VGGs. The reason for this difference needs a further investigation. Lastly, the presence or absence of skip connections does not alter the degradation of out-of-distribution performance. However, in the absence of skip connections, the deterioration is more severe, aligning with the observation that it correlates with the numerical rank of the representations.

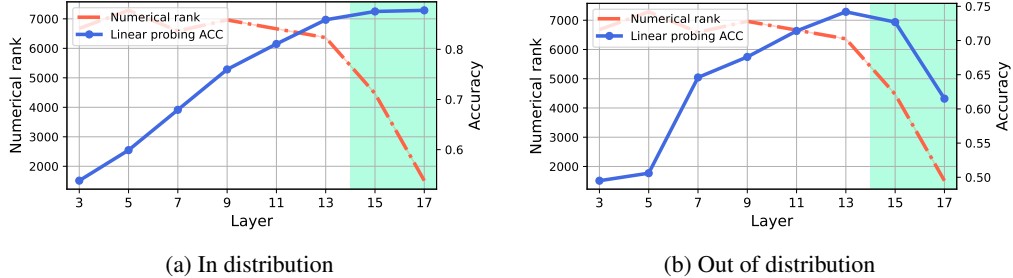

(a) In distribution

(b) Out of distribution

Figure 52: In and out of distribution linear probing performance for ResNet-18 without skip connections trained on CIFAR-10. The shaded area depicts the tunnel, the red dashed line depicts the numerical rank and the blue curve depicts linear probing accuracy (in and out of distribution) respectively. Out-of-distribution performance is computed with random 10 class subsets of CIFAR-100.

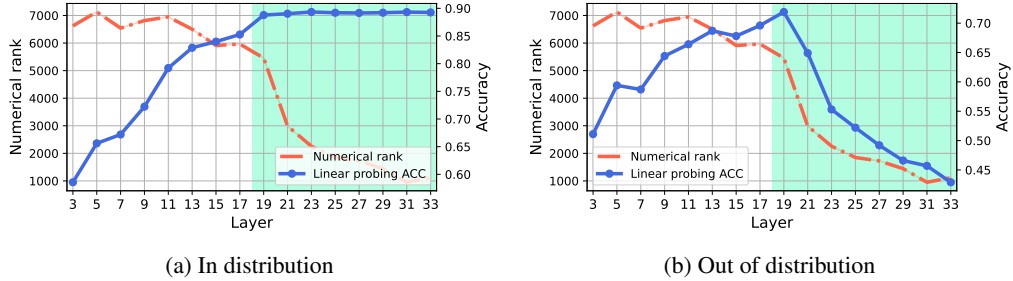

(a) In distribution

(b) Out of distribution

Figure 53: In and out of distribution linear probing performance for ResNet-34 without skip connections trained on CIFAR-10. The shaded area depicts the tunnel, the red dashed line depicts the numerical rank and the blue curve depicts linear probing accuracy (in and out of distribution) respectively. Out-of-distribution performance is computed with random 10 class subsets of CIFAR-100.

