# OpenReview forum: "The Tunnel Effect: Building Data Representations in Deep Neural Networks"
_NeurIPS.cc/2023/Conference — NeurIPS 2023 poster_

### Official Review · Reviewer_Kpac · 2023-06-21

**Soundness:** 2 fair
**Presentation:** 3 good
**Contribution:** 2 fair
**Rating:** 6
**Confidence:** 4

**Summary:**

The paper offers an empirical study of deep neural networks. The focus is on the role of intermediate layers in building a representation that is linearly separable and can eventually solve the task. The work highlights the fact that this linearly separable, low-rank representation emerges at a depth that is a fraction of the total depth (usually 1/3 to 2/3 for common image classification setups). Layers before such point are named "the extractor", and layers afterward "the tunnel". The authors further investigate the role of "the tunnel effect" on transfer learning and catastrophic forgetting.

**Strengths:**

The main strength of the paper is the thoroughness of the empirical study. Although only pertaining to image classification (as mentioned by the authors), the study includes a wide range of benchmark datasets and architectures, and shows results with plots that are easy to parse.
Connecting the tunneling effect to practical recommendations for transfer learning is another interesting point.

**Weaknesses:**

Although the authors put a strong emphasis on the novelty of their results, the fact that a linearly separable representation emerges well before the final layer is not completely novel. See e.g. [1] for an analysis using similar methods.
Also, the reasons behind the so-called paradox that motivates the work are not completely clear. It's not evident how the fact that capacity increases with depth, is at stake with the fact that there is an "inductive bias" toward simple solutions.
In my view, the motivation paragraph in the introduction would need to be clarified, and the conclusion should clearly state how this paradox has been addressed by the paper.

[1] Feature learning in deep classifiers through Intermediate Neural Collapse, Rangamani, Akshay; Lindegaard, Marius; Galanti, Tomer; Poggio, Tomaso.

**Questions:**

- Section 3.1 put emphasis on the fact that the tunneling effect emerges "early" during training. How do you assess what early is? Is the effect happening before/same time/after train and test losses start decreasing? It would be nice to have a comparison of these different timescales to better assess the claim of the section.
- In the introduction, the authors claim to challenge the idea that deeper layer capture more complex and task-specific features. What are the results that question this idea? For example, the observation that the transfer learning performance deteriorates after neural collapse (i.e. in the tunnel part) is consistent with the idea that the representation in these layers becomes task-specific. The fact that more complex tasks require a longer "extractor" is consistent with the idea that more complex tasks require more complex representations to be built and more complex representations are built in a hierarchical manner through more layers.
- I find the results of Figure 9 exemplificative of the fact that the network uses always the first layers to solve the task, independently of its total depth. It would be nice to have the same Figure for a CNN.
- Minor: What is the difference between blue and orange dots in Figure 10 (top).

**Limitations:**

The authors clearly highlight that their work focuses only on image classification and that a study for other data modalities would be interesting. I think this is a fair limitation and does not impact my evaluation of their work.
Also, at first sight, the role of skip connection would be very relevant for the effect studied in this work, and possibly it would deserve to be discussed more in the main text.

---

> ### Author Rebuttal · Authors · 2023-08-09
>
> Thank you for your thorough review.
>
> **The fact that a linearly separable representation emerges well before the final layer is not completely novel.**
>
> We do agree that this observation has been made in multiple works, including those we cite. We consider the primary value of our work to be *conceptualization and providing a coherent perspective of how representations are formed*. In particular, we do not mean to claim that all observations are new. We instead attempt to build standalone empirical evidence to support our tunnel hypothesis and explore its consequences in transfer learning or continual learning scenarios. We have added the reference to the Related Works section:
>
> >Several recent works~\cite{ansuini2019intrinsic,li2022principled,rangamani2023feature} have extended the observation of NC and explored its impact on different layers, with a notable emphasis on deeper layers. \cite{li2022principled} establishes a link between collapsed features and transferability. In our experiments, we delve into tunnel creation, analyzing weight changes and model behavior in a continual learning scenario, revealing the task-agnostic nature of the tunnel layers.
>
> **Clarification of the so-called paradox.**
>
> We acknowledge that using the term “paradox” may have caused confusion, and we have revised the introduction and conclusion to eliminate any ambiguity. The revised paragraph in the introduction now reads as follows:
>
> The revised paragraph in the introduction now reads as follows:
>
> > Extensive research has ..... to simplify representations with increasing depth [38,7]. This phenomenon arises because, despite their large capacity, these networks strive to compress hidden representations by focusing on discriminative patterns during supervised training [38, 7, 8, 30]. Motivated by these findings, we aim to investigate this phenomenon further and build a comprehensive picture to answer the following research question:
> >
> > "How do representations depend on the depth of a layer?"
>
>
> We add the following paragraph to the conclusions:
>
> > We emphasize that our motivation for investigating this (i.e., tunnel) phenomenon is to build a coherent picture encompassing our experiments and evidence in the literature. Specifically, we aim to understand better how the neural networks handle the representation-building process in the context of depth.
>
> **In the introduction, the authors claim to challenge the idea that deeper layer capture more complex and task-specific features. What are the results that question this idea?**
>
>
> We recall the following facts from our study
>
> 1. Experiments show that the tunnel layers, with decreased out-of-distribution (OOD) performance, have minimal contribution to in-domain (ID) performance (Figure 1, 2, 7). [roughly speaking the representations 'do not change' in the tunnel as it comes to the performance]
> 2. The tunnel layers' representations are identical (measured via CKA), indicating a lack of meaningful task-specific features (Figure 4) [roughly speking, the representations do not change in the tunnel when measure with CKA]
> 3. Experiments demonstrate that the tunnel remains unchanged when optimizing the network for a new task, allowing for transferability across tasks without retraining (Table 4). [i.e. the tunnel is task-agnostic]
>
> Perhaps the current phrasing is somewhat misleading. Would you find the following better:
> > The deeper layers (i.e. the tunnel) concentrates on compressing the representations instead of building more complex ones.
>
>
>
> **The role of skip connection would be very relevant and deserve to be discussed more in the main text.**
>
> We agree that skip-connections seem to play a very relevant role in this phenomenon. In the updated version of the paper, we included the study about the role of skip connections in the main text.
>
>
>
> **The early emergence of the tunnel**
>
> The tunnel effect develops gradually, making it difficult to precisely identify the specific step or epoch when it becomes noticeable from our standpoint. However, regarding rank, we can observe a sudden decrease within the initial training steps, as depicted in Figure 6. To gain further insights, we conducted additional experiments using linear probes attached to the network at various training stages. The results in Figure 4 of the attached PDF indicate that the accuracy saturates in deeper layers as early as the 10th epoch despite an overall gradual improvement in accuracy in later epochs. These findings suggest tunnel formation occurs within 5% of the training process.
>
>
>
> **I find the results of Figure 9 exemplificative of the fact that the network uses always the first layers to solve the task (...). It would be nice to have the same Figure for a CNN.**
>
> We provide the analogous Figure for the VGG-based networks of various depths -- see Figure 2 in the attached PDF file.
>
>
> **Minor: What is the difference between blue and orange dots in Figure 10 (top).**
>
> In this scenario each network (of various depth) was trained on a sequence of two tasks. The blue and orange colors depict the performance of the models after training on the first task and second task respectively. We'll update the Figure to include this information in the legend and caption.
>
>
>
>
> If your concerns have been sufficiently addressed in our responses, we humbly seek your support for the paper and ask you to consider improving your score. Should you have any further concerns or additional points to raise, we are eager to address them.

---

> > ### Comment · Reviewer_Kpac · 2023-08-11
> >
> > I thank the authors for their answers.
> >
> > The picture that emerges from this work seems to suggest that, in the first part of the network, deeper layers capture more complex and task-specific features, while this is not the case for the tunnel part, which is mainly used to compress the representation and is not crucial for accuracy.
> >
> > I raise my rating to 6.

---

> > > ### Author Response · Authors · 2023-08-14
> > >
> > > Again, we thank the reviewer for helping us raise this paper's quality.
> > > Should you have any other questions, we are happy to answer them.

---

### Official Review · Reviewer_EeeK · 2023-07-05

**Soundness:** 3 good
**Presentation:** 4 excellent
**Contribution:** 4 excellent
**Rating:** 7
**Confidence:** 4

**Summary:**

The tunnel effect is described for deep overparameterized networks, whereby early layers form a linearly separable representation while later layers form a "tunnel" which passes this representation to the output without substantial change, other than compression (reducing its rank, a.k.a. discarding information). A large number of experiments show that this effect occurs in a variety of different models, datasets, and training times, and impacts out-of-distribution and continual learning settings. The effect implies that the capacity of a given architecture/dataset is fixed.

**Strengths:**

This paper provides a clear and comprehensive argument for both the existence of its hypothesis and its effects on several relevant subdomains of DNN research. The experiments are thorough, well organized, and address many potential concerns (such as consistency of the observations across different models and datasets). Some subtle details are handled correctly (e.g. use of unbiased CKA estimator in appendix E). The effect is convincing and could point to significant improvements in how transfer and continual learning is handled.

**Weaknesses:**

The procedure for computing the numerical rank should be given fully (e.g. the value of the threshold $\sigma$ is not explicitly stated). In particular, figure 6 indicates some issues (see Questions below). Also, figure 6 is missing shading for the tunnel region after training, which would help readers reference other figures.

The difference in weights in figure 5 is not fully explained (e.g. $\tau$ should be defined). Also, while the difference is normalized relative to $\theta^0$ (the initial weight norm?, or is it the number of weights?), a better comparison would be relative to the norm of the mean change iterations/epochs, since the learning rate changes over training. As a result, the experiment cannot rule out the possibility that the tunnel is changing relative to other layers after the learning rate is reduced.

Overall the evidence from CKA is lacking (figure 4) as it is only shown for a MLP. However, this is not a central piece of evidence.

**Questions:**

In figure 6, why is the numerical rank for higher layers already less than that of lower layers at initialization? Given that random matrices tend to be near full-rank, this points to an issue with how numerical rank is computed (e.g. the threshold for determining rank should be layer-dependent, since different layers have different weight scales).

**Limitations:**

ImageNet-scale networks and Transformer architectures are out of scope of this work. But given the importance of such networks to transfer learning applications, the obvious next step is to investigate the tunnel effect in these networks. Unlike the CIFAR/CINIC-scale models in this work, it is unclear what degree of overparameterization is occurring in ImageNet-scale networks and Transformers.

---

> ### Author Rebuttal · Authors · 2023-08-09
>
> Thank you for the effort you put into reviewing our work. We find your feedback valuable and helpful in improving the quality of the paper!
>
> **The procedure for computing the numerical rank should be given fully.**
>
> We added the following clarification in the text:
> > The threshold $\sigma$ is set to $\sigma_{1}*1e-3$, where $\sigma_{1}$ is the highest singular value of the particular matrix.
>
>
> **Also, Figure 6 is missing shading for the tunnel region after training, which would help readers reference other figures.**
>
> We added shading in Figure 6.
>
> **The difference in weights in figure 5 is not fully explained (e.g. should be defined).**
>
> We fixed the notation in the caption of Figure 5 and explained the formula in more detail.
>
> >The tunnel layers stabilize early during training. Color depicts the norm of weights difference between subsequent checkpoints (rows) for a given layer (columns). Norm is computed with $\frac{1}{\sqrt{n m}}\left\|\theta_d^{\tau_1}-\theta_d^{\tau_2}\right\|_{2^{\prime}}$ where $\theta_d^\tau \in \mathbb{R}^{n m}$ is flattened matrix of weights from $d$-th layer at checkpoint $\tau$. The values are clipped at $0.02$ for better presentation. At epochs 80 and 120, the learning rate is decayed. VGG-19, CIFAR-10.
>
> **The experiment cannot rule out the possibility that the tunnel is changing relative to other layers after the learning rate is reduced.**
>
> Thank you for pointing out that possibility. To check whether tunnel formation is reversed after decaying the learning rate, we split Figure 5 into three parts, with the scale adjusted by the decay factor. The results in Figure 1 in the attached PDF align with the previously presented material. Despite the decaying learning rate, the model's behavior is consistent throughout the training, and the split between the extractor/tunnel part is visible in all phases. We will update the text and Figure 5 accordingly in the paper.
>
> **The CKA evidence is limited to an MLP in Figure 4, which is not a pivotal part of the argument.**
>
> In Figure 3 in the attached PDF, we present the plots for CKA similarities for VGG-19 and ResNet-34 (we also added the Figure to the paper). VGG-19 exhibits similar representations for deeper layers (starting around layer 8). In the case of ResNets, the pattern is less visible. We suspect this can be caused by skip-connections and their impact on the final representations, as mentioned in the last paragraph of the Limitations section.
>
> **In Figure 6, why is the numerical rank for higher layers already less than that of lower layers at initialization?**
>
> As we explained earlier, the threshold $\sigma$ used for estimating the numerical rank is computed separately for each layer based on their top-singular value making it layer-dependent. Also, please note that the dimensionalities of the lower-layer representations are higher. Due to computational constraints, we subsample these layers to have at most 8000 features (e.g. [1]). Yet, the effect of decreased rank in upper layers is also present in MLPs where the dimensionality of the representation is constant across layers, and we do not subsample representations.
>
> [1] G. Alain, Y. Bengio - Understanding intermediate layers using linear classifier probes
>
> If your concerns have been sufficiently addressed in our responses, we humbly seek your support for the paper.

---

> > ### Comment · Reviewer_EeeK · 2023-08-16
> >
> > Thank you for the clarifications. After reading the other comments, I believe this paper provides solid evidence but some of its observations are known in the literature. Nevertheless the original method of replication is valuable, and I will keep my score.

---

> > > ### Author Response · Authors · 2023-08-17
> > > **Thank you**
> > >
> > > Thank you again for your work towards making our paper better.

---

### Official Review · Reviewer_dmw5 · 2023-07-06

**Soundness:** 2 fair
**Presentation:** 2 fair
**Contribution:** 1 poor
**Rating:** 3
**Confidence:** 4

**Summary:**

This paper proposes the Tunnel Hypothesis: Training of deep networks splits layers into two distinct phases: (1) extractor phase and (2) tunnel phase. Extractor phase learns the linearly separable features whereas the tunnel phase compresses the representations. The authors provide evidence towards degrading effects of this tunnel effect on OOD (out-of-distribution) samples. Further, better understanding of continual learning may be possible due to the proposed hypothesis.

**Strengths:**

The paper has following strengths:

1. The OOD representation section was very interesting.

2. Applications to continual learning based on knowledge gained from the detailed analysis from this paper can also be useful.

3. Overall, a lot of work has gone into this paper (many experiments).


**Weaknesses:**

The paper has following weaknesses:

1. Many of the observations are not particularly surprising. I think a lot depends on capacity of the network and the difficulty of the task at hand. It is not surprising that for a given task and a type of network, representations get learned up to a certain layer and then the remaining layers simply make the representations more compact. Indeed, if the task becomes more complex or more difficult, the tunnel length would reduce (since more layers would be spent trying to learn more complex features). This is clear from Table 1 where for ResNet-34, the tunnel length significantly reduces when going from CIFAR-10 to CIFAR-100 (from 20 to 29 layers). Thus, when the task became more difficult, more layers started getting used to learn better features. Similarly, for simple MLPs, we know that beyond a certain depth, depth does not help (this comes from many other studies, e.g., that analyze gradient properties, etc.: beyond a certain depth and without skip connections, adding more layers does not help due to vanishing gradients). The insight that “many later layers do not contribute significantly to accuracy” is also known and is precisely why “deep network pruning” literature is not able to prune later layers too significantly.

2. In the introduction section, the authors claim that “they challenge the commonly held intuition that deeper layers are responsible for capturing more complex and task-specific features”. I do not see any evidence that they changed this commonly held view. In fact, many of their experiments reinforce exactly the common viewpoint. Specifically, the authors show that later layers hurt the OOD performance. This indicates that the later layers got specialized towards the within-distribution task which is why they hurt the OOD task. Hence, if the commonly held view is being reinforced with the evidence provided by the authors, there is nothing particularly surprising about the findings.

3. I think the proposed work can have significant value in the field of continual learning and also multi-task/multi-modal learning if the observed insights can be used to guide novel architectural designs and/or loss functions. Unfortunately, the current work (despite a lot of hard work) only plays around with toy datasets in that problem space. If the authors can build further on the new insights and create new models/losses for the aforementioned areas, this can be an impactful work.


**Questions:**

Please see above.

**Limitations:**

Yes.

---

> ### Author Rebuttal · Authors · 2023-08-09
>
> We want to thank the reviewer for their valuable feedback.
>
> We acknowledge some presentation issues, which we discuss in the general answer. Please let us know if you find it satisfactory. We discuss the other issues below.
>
>
> **Increasing depth does not help because representations are learned up to a certain layer and subsequent layers make them compact. (Weakness #1)**
>
> We have thoroughly researched existing literature but have yet to find any work that describes this particular behavior. However, we are open to any references or sources you might be aware of in this context, and we would be grateful for any insights you can provide.
>
> Furthermore, credible alternatives, without a tunnel, could be easily imagined. For example, considering the perspective of gradient flow, one would anticipate a contrary pattern. As the gradient norm diminishes when we move away from the output layer, one could conjecture that the upper layers should "learn more" than the bottom ones.
>
> Last, the contribution of this work lies in conceptualizing the tunnel hypothesis underpinning several observations made in referenced literature and extensively examining its implications in transfer learning and continual learning settings.
>
>
>
> **Lack of support for challenging the view that deep layers learn task-specific features. (Weakness #2)**
>
> The scattered presentation could have harmed clarity and readability. Thus, we provide a concise summary of these results in one place:
>
> 1. The tunnel layers, which demonstrate a decrease in out-of-distribution (OOD) performance, do **not** contribute significantly to the in-distribution (ID) performance (Figure 1, 2, 7).
> 2. We want to emphasize that the tunnel layers remain unchanged when optimizing the network for a novel task (see Table 4). This transferability across tasks (without tunnel retraining) falsifies the task-specific view of the deeper layers.
> 3. Measured with CKA, representations at the tunnel's beginning and end are roughly the same. This suggests that no significant transformation was applied to them in the tunnel layers (Figure 4).
>
> We are not aware of any such findings being comprehensively reported in the literature. Thus, our research provides a new, structured, and arguably valuable perspective. We'd be happy to engage in further discussion or amend our work with prior work if the reviewer offers any specific references.
>
>
> **If the authors can build further on the new insights and create new models/losses for the aforementioned areas, this can be an impactful work. (Weakness #3)**
>
> We agree this is an exciting area for further research. We have compiled a paragraph of recommendations based on new insights from the tunnel hypothesis (see below). We include these recommendations in the conclusions section.
>
> > Our tunnel hypothesis has multiple consequences relevant to downstream tasks.
> >
> > In particular, focusing on the tunnel entry features is promising when dealing with distribution shift due to its strong performance with OOD data.
> For continual learning, regularizing the extractor should be enough, as the tunnel part exhibits task-agnostic behavior. Skipping feature replays in deeper layers or opting for a compact model without a tunnel can combat forgetting and enhance knowledge retention.
> >
> >For efficient inference, excluding tunnel layers during prediction substantially cuts computation time while preserving model accuracy, offering a practical solution for resource-constrained situations.
>
>
> **Indeed, if the task becomes more complex or more difficult, the tunnel length would reduce (since more layers would be spent trying to learn more complex features).**
>
> This observation is indeed compelling and is further explored in section 3.3 (Table 3), where we confirm the intuition that training on more complex datasets results in shorter tunnels. However, the complexity of the dataset is understood here as the number of classes. Note that more complex datasets regarding the number of samples do not impact the tunnel's length (compare CIFAR-10 vs CINIC-10 in Table 1). Formalizing this relationship with more rigorous mathematical treatment is an exciting direction for future work.
>
>
> **The insight that “many later layers do not contribute significantly to accuracy” is also known and is precisely why “deep network pruning” literature is not able to prune later layers too significantly.**
>
> Your comment suggests a link between layer depth and pruning potential, but we're unsure how this aligns with our findings and whether we should reference any works that confirm those observations in our paper. We'd be happy to see a clarification of this remark and amend our submission accordingly.
>
>
> We again thank the reviewer for raising important issues. We hope that our answers are satisfactory. If not, we'd be happy to provide more details. Otherwise, we'd appreciate if the reviewer reconsidered the final score of our submission.

---

> > ### Comment · Reviewer_dmw5 · 2023-08-19
> > **Thanks for the response**
> >
> > I thank the authors for the response. As I said in the original review, I do find their OOD related findings interesting. Sorry for not providing concrete references in my review.
> >
> > Here are four references that may have made me think that the observations were not too surprising:
> > 1. Pruning literature [Neurips 2019]: https://arxiv.org/pdf/1906.02773.pdf. See Fig. 1 and description in section 3.1 (global pruning paragraph) in this paper. It says that later layers can be pruned more. Hence, I thought that it was known knowledge that later layers have more redundant knowledge that can be removed. It is also known that if a network is too compact, it is hard to prune such networks (as seen by practical difficulty in pruning MobileNets, EfficientNets, etc.). Thus, if the task complexity is too much for the model at hand, many layers would play important role (and thus the tunnel would reduce). The tunnel hypothesis is essentially stating the same thing -- that the later layers do not contribute much new information if a task is relatively simple (thus a longer tunnel). If a task becomes more difficult, the tunnel reduces.
> >
> > 2. Deep Equilibrium Models (DEQ) [Neurips 2019]: https://arxiv.org/pdf/1909.01377.pdf. The whole motivation behind these networks was that after a certain layer, representation does not change much (i.e., they approach equilibrium). This can again be seen as an evidence towards representations not changing much after a certain number of layers. Please also check some other references on page 1 of the original DEQ paper above (refs [18,8,15]).
> >
> > 3. Feature visualization kinds of research [Distill 2017]: https://distill.pub/2017/feature-visualization. These are just some papers that talk in detail about how representations form (but from visualization perspective). This one may not have too much insight but it shows empirically how representations form.
> >
> > 4. How information content changes through the training: https://arxiv.org/pdf/1703.00810.pdf. This formally looks into how representations compress over the course of training using mutual information measures, etc. There was a lot of debate around this work, so I encourage authors to look at other follow up works also (e.g., ICLR 2018 https://openreview.net/forum?id=ry_WPG-A-).
> >
> > On the "more complex tasks requiring shorter tunnels" section above: I was only saying that given that features form slowly through the network layers (edges to complex shapes, as shown in the above Distill 2017 visualization paper), I imagined that for a more complex task, the network would spend more layers trying to learn the representations (and thus the tunnel would be shorter). And if the task is easy, several layers would learn redundant info (and, hence, layer output would not change between such layers...which is clear from your CKA analysis). Moreover, given the pruning literature, I always assumed this redundancy was happening towards the end. Maybe it is just me...but these are some of the reasons I was not surprised.
> >
> > If this paper had more theoretical results, that would make the case much stronger. For now, I will keep my rating. I will leave it up to other reviewers and AC in case I genuinely missed something.

---

> > > ### Author Response · Authors · 2023-08-20
> > > **Thank you**
> > >
> > > First, we’d like to say that we enjoy the discussion with the reviewer, even if we disagree. Thank you for raising relevant points.
> > >
> > >
> > > We admit some deficiencies in presentation, particularly the emphasis on 'surprisingness,' which is now removed. We have amended the paper as outlined in the general answer 'scope and novelty' section. Now, we underline that the value of our work is *conceptualization, providing a coherent picture, and studying its implications*. We believe it is valuable, and for example, it constitutes the solid ground for building further insights, e.g., the mentioned OOD results. Please let us know if any more changes would be beneficial.
> > >
> > > **Regarding the papers**, we thank you for suggesting them. The following paragraph will be added to the related work section:
> > > > The analysis of representations across layers has been a focal point in many related studies. Visualization of layer representations indicates that higher layers capture intricate and meaningful features, often formed through combinations of lower-layer features~[7]. This phenomenon potentially accounts for the extension of feature extractors for complex tasks. Work [1] builds a theoretical picture that stacked sequence models tend to converge to a fixed state with infinite depth and proposes a method to compute the finite equivalent of such networks. The framework of [1] encompasses previous empirical findings of  [2,3,4]. Independently, research on pruning methods has highlighted a greater neuron count in pruned final layers compared to initial layers [8], which is in line with the tunnel's existence. Furthermore, in [5,6], authors showed that training neural networks may lead to compressing information contained in consecutive hidden layers.
> > >
> > > In more detail:
> > >
> > > Referenced articles [2,3,4] show the near-state-of-the-art performance of weight tying in deep sequence models. We believe that these findings are orthogonal to our extractor tunnel hypothesis. Deep Equilibrium Models [1] extend these results and present a very interesting analysis, which states deep sequence models tend to converge to a steady state. The first significant difference is the assumption of homogeneity, i.e., the states are transformed iteratively using the same function instead of ‘different’ layers. Secondly, [1] is more on the theoretical side; we show empirically that the phenomena in question occur for ‘standard-depth’ neural nets. Thirdly, we show that the representations are not static in the tunnel when we probe them with the OOD tasks.
> > >
> > > Regarding the Information Bottleneck concept [5], it is compatible with our observation. However, we do not see a direct way of deriving the extractor-tunnel transition and its relative sharpness. Moreover, [5] hinges heavily on the particular activation function, as shown in [6], and does not necessarily hold for the RELU activations used in our work. Finally, again [5] does not allow to derive the OOD behavior.
> > >
> > > The visualizations presented in [7] nicely illustrate the extractor, although, in our opinion, they do not suffice to directly derive the existence of the tunnel nor the sharp transition.
> > >
> > > We find the pruning results somewhat orthogonal. We would say that our results and [8] complement each other rather than can be derived from each other.
> > >
> > > Last but not least, we empathize with the reviewer's sentiment that these works are mounting evidence for the extractor tunnel phenomenon. Nevertheless, in our view, pinpointing and naming it is of value to the community.
> > >
> > > **If this paper had more theoretical results, that would make the case much stronger.**
> > >
> > > We agree; we consider this work as a solid starting ground for future theoretical research. It has been added to the limitations and further work section.
> > >
> > >
> > > [1] Bai, S., Kolter, J. Z., & Koltun, V. (2019). Deep equilibrium models. Advances in Neural Information Processing Systems, 32.
> > >
> > > [2] Dabre, R., & Fujita, A. (2019, July). Recurrent stacking of layers for compact neural machine translation models. In Proceedings of the AAAI Conference on Artificial Intelligence (Vol. 33, No. 01, pp. 6292-6299).
> > >
> > > [3] Bai, S., Kolter, J. Z., & Koltun, V. (2018). Trellis networks for sequence modeling. arXiv preprint arXiv:1810.06682.
> > >
> > > [4] Dehghani, M., Gouws, S., Vinyals, O., Uszkoreit, J., & Kaiser, Ł. (2018). Universal transformers. arXiv preprint arXiv:1807.03819.
> > >
> > > [5]  Shwartz-Ziv, R., & Tishby, N. (2017). Opening the black box of deep neural networks via information. arXiv preprint arXiv:1703.00810.
> > >
> > > [6] Saxe, A. M., et al., (2019). On the information bottleneck theory of deep learning. Journal of Statistical Mechanics: Theory and Experiment, 2019(12), 124020.
> > >
> > > [7] Olah, C., Mordvintsev, A., & Schubert, L. (2017). Feature visualization. Distill, 2(11), e7.
> > >
> > > [8] Morcos, A., et al.,  (2019). One ticket to win them all: generalizing lottery ticket initializations across datasets and optimizers. Advances in neural information processing systems, 32.

---

### Official Review · Reviewer_xLcw · 2023-07-07

**Soundness:** 3 good
**Presentation:** 3 good
**Contribution:** 3 good
**Rating:** 6
**Confidence:** 4

**Summary:**

This paper shows an effect of deep neural networks when trained for classification tasks — the initial layers create linearly separable features, and the later layers collapse the features for the final prediction. This phenomenon is explored with extensive experiments.

**Strengths:**

- The paper explored a very interesting phenomenon of how the features are learned dynamically through layers.

- The paper performed extensive experiments to show how this "tunnel effect" affects the model performance under different settings.

- The experiments are well-designed, and the results are demonstrated well.




**Weaknesses:**

-  This is not the first/only paper that discovered some similar effects under a similar setting, therefore, a more comprehensive comparison with them in the related work section as well as a clarification on the contribution should be added. e.g. [1] examines the feature "intrinsic dimension", [2]  analyzes the generalization effects of feature neural collapse on in-domain and out-of-domain data.

- The authors mentioned the network can be split into the extractor and the tunnel which compress the features, however, it can be difficult to systematically split the network, it seems the author also did not provide a systematically split of the network based on the numerical rank.


[1] Ansuini et. al. Intrinsic dimension of data representations in deep neural networks.
[2] Li et. al. Principled and Efficient Transfer Learning of Deep Models via Neural Collapse.


**Questions:**

- Can the tunnel effect, or the numerical rank provide practical guidance on which feature we should use for different problems, e.g. out-of-distribution data, continual learning?

- Following up on the previous question, if this tunnel effect can provide guidance, how can we use this tunnal effect while model training and/or model inference.




**Limitations:**

The limitations of the paper is relatively thoroughly discussed.

---

> ### Author Rebuttal · Authors · 2023-08-09
>
> We thank the reviewer for their comments.
>
> **Considering similar findings in other papers, enhance the related work section with comparisons and clarify contributions.**
>
> We have added the references [1] and [2] to the related works section:
>
> >Several recent works~\cite{ansuini2019intrinsic,li2022principled,rangamani2023feature} have extended the observation of NC and explored its impact on different layers, with a notable emphasis on deeper layers. \cite{li2022principled} establishes a link between collapsed features and transferability. In our experiments, we delve into tunnel creation, analyzing weight changes and model behavior in a continual learning scenario, revealing the task-agnostic nature of the tunnel layers.
>
> **Systematically dividing the network is challenging, and the authors did not offer a numerical rank-based network split.**
>
> We are still determining if we understand this concern correctly (if the following answer is off, please let us know). Our experiments observed relatively straightforward patterns, where the accuracy flattens (e.g.,> 90% of the final value) coincides with the point where the rank starts to drop. This gives a rather sharp boundary between the extractor and the tunnel.
>
>
> **Can the tunnel effect, or the numerical rank provide practical guidance on which feature we should use for different problems, e.g. out-of-distribution data, continual learning?**
>
> We have compiled a paragraph of recommendations based on new insights from the tunnel hypothesis (see below). We include these recommendations in the conclusions section.
>
> > Our tunnel hypothesis has multiple consequences relevant to downstream tasks.
> >
> > In particular, focusing on the tunnel entry features is promising when dealing with distribution shift due to its strong performance with OOD data.
> For continual learning, regularizing the extractor should be enough, as the tunnel part exhibits task-agnostic behavior. Skipping feature replays in deeper layers or opting for a compact model without a tunnel can combat forgetting and enhance knowledge retention.
> >
> >For efficient inference, excluding tunnel layers during prediction substantially cuts computation time while preserving model accuracy, offering a practical solution for resource-constrained situations.
>
>
> **How can we use this tunnal effect while model training and/or model inference?**
>
> One practical approach is to conduct model inference without the tunnel layers. As these layers do not contribute to the final performance. In continual learning, if tasks have the same class count, the tunnel remains unchanged; thus, not updating or removing the tunnel layers can be considered for new tasks.
>
>
> If our responses have adequately addressed your concerns, we kindly request your support and considerating of improving your score. If you have any further concerns or additional points to raise, we are eager to address them. Your insights are valuable in enhancing the quality and impact of our research.

---

> > ### Comment · Reviewer_xLcw · 2023-08-16
> >
> > Thanks to the authors for the rebuttal. I think the author successfully addressed my concerns and solved my questions. The paper is very interesting, therefore I will keep my original score and suggest accepting the paper.

---

> > > ### Author Response · Authors · 2023-08-16
> > > **Thank you**
> > >
> > > Thank you for supporting our work and, again, for useful comments and suggestions.

---

### Author Rebuttal · Authors · 2023-08-09

## General Response
Dear reviewers,

Many thanks for providing valuable feedback in your reviews, both positive and negative. We are delighted to note that all the reviewers prized the scale of our experimentation and found our results very interesting (xLcw, dmw5), with xLcw reporting that they are 'well-designed' and EeeK agreeing on their thoroughness and organization. Moreover, dmw5, Eeek, and Kpac indicated that the presented analysis could have implications for downstream tasks like OOD or continual learning. The reviewers expressed interest in further pursuing this line of research in other modalities (Kpac).

### Scope and novelty
On the negative side, reviewers dmw5 and Kpac raised concerns regarding the novelty of our findings and, consequently, the contribution of our work. We consider the primary value of our work to be *conceptualization and providing a coherent perspective of how representations are formed*. In particular, we do not mean to claim that all observations are new. We instead attempt to build a standalone empirical evidence to support our tunnel hypothesis. Moreover, we acknowledge that we missed discussing more thoroughly [1,2,3], which will be added.


To be clear, we apply the following amendments to the paper:
- we add the papers (see above) to the related work section along with the discussion
- we update the motivation and conclusion paragraphs (see below)
- we change the first point of the contribution to "We *conceptualize* and extensively examine the tunnel effect, namely, deep networks naturally split into the extractor responsible for building."


**The revised paragraph in the introduction now reads as follows:**

> Extensive research has ..... to simplify representations with increasing depth [38,7]. This phenomenon arises because, despite their large capacity, these networks strive to compress hidden representations by focusing on discriminative patterns during supervised training [38, 7, 8, 30]. Motivated by these findings, we aim to investigate this phenomenon further and build a comprehensive picture to answer the following research question:
>
> "How do representations depend on the depth of a layer?"


**We add the following paragraph to the conclusions:**

> We emphasize that our motivation for investigating this (i.e., tunnel) phenomenon is to build a coherent picture encompassing our experiments and evidence in the literature. Specifically, we aim to understand better how the neural networks handle the representation-building process in the context of depth.

[1] Ansuini et. al. Intrinsic dimension of data representations in deep neural networks.

[2] Li et. al. Principled and Efficient Transfer Learning of Deep Models via Neural Collapse.

[3] Feature learning in deep classifiers through Intermediate Neural Collapse, Rangamani, Akshay; Lindegaard, Marius; Galanti, Tomer; Poggio, Tomaso.

---

### Decision · Program_Chairs · 2023-09-21

**Decision:**

Accept (poster)

**Comment:**

The paper received mixed recommendations after the rebuttal. The majority of reviewers (3 out of 4) find the paper has value in conceptualizing and experimentally validating the "tunnel effect". On the other hand, reviewer dmw5 maintains their rejection rating, concerning novelty and the additional value given certain evidences from related research works. The AC checked all the materials and sides with the majority vote, especially given the OOD study (which dmw5 also finds interesting) and the continued effort that improves the positioning and presentation of the paper. Please incorporate necessary changes in the final version.